# Positive affect modulates memory by regulating the influence of reward prediction errors
Salman E. Qasim [1,2], Aarushi Deswal[3], Ignacio Saez [2,4,5,6] & Xiaosi Gu [1,2,4] ✉

How our decisions impact our memories is not well understood. Reward prediction errors (RPEs), the difference between expected and obtained reward, help us learn to make optimal decisions-providing a signal that may influence subsequent memory. To measure this influence and how it might go awry in mood disorders, we recruited a large cohort of human participants to perform a decision-making task in which perceptually memorable stimuli were associated with probabilistic rewards, followed by a recognition test for those stimuli. Computational modeling revealed that positive RPEs enhanced both the accuracy of memory and the temporal efficiency of memory search, beyond the contribution of perceptual information. Critically, positive affect upregulated the beneficial effect of RPEs on memory. These findings demonstrate how affect selectively regulates the impact of RPEs on memory, providing a computational mechanism for biased memory in mood disorders.

While making decisions, we often try to predict the consequences of our choices so that we can pick the best option. Very often, the outcomes of our choices do not match with our predictions, generating signals called reward prediction errors (RPEs). Reinforcement learning (RL) models capture this framework of trial-by-trial learning, which is critical for understanding decision-making in animals and humans and for building artificial agents[1,2]. However, it is not well understood whether or how, RPEs that are central to RL models may have a more lasting effect on cognition. For example, if you choose a restaurant for dinner and unexpectedly find $100 on the ground while eating there, does that dining experience stand out more in your memory than other restaurants? Here, we examine how reinforcement learning, driven by RPEs, influences the encoding and retrieval of memories related to prior decisions.

Neurally, RPEs have been associated with phasic dopamine release in the brain[3], which are then used to guide how we evaluate and optimize subsequent choices. Understanding how RPEs impact memory is important because RL algorithms provide biologically plausible models of dopamine-driven learning[1,4] facilitated by mesocortical and cortico-striatal circuits for habits, action selection, and decision-making. As a result, one prominent theory of RPE-mediated memory is that midbrain dopaminergic release strengthens memory encoding and consolidation[5]. Indeed, recent behavioral evidence suggests that rewards[6–8] and RPEs[9–13] imbue stimuli with subjective salience that modulates memory[14]. However, the specific direction, timing, and magnitude of this effect vary across studies[14] suggesting the

involvement of alternative, unaccounted-for sources of prioritized memory. Most prominent among these is the contribution that perceptual information makes to mnemonic salience, independent from the reward outcomes associated with the stimuli[15,16]. Perceptual memorability is not explained by low-level visual salience, cognitive control, or priming[15], and represents a parallel path to enhanced memory that has distinct neural circuits and behavioral implications. Therefore, in order to understand how decision-making processes shape our memories for those decisions, it is critical to understand the interplay between stimulus-specific perceptual memorability and reward prediction during memory processes[17–19].

Untangling the distinct contributions of perceptual and reward information to memory is also important to understanding how disruptions to affective states—such as depressive and anxious mood—affect the relationship between RPEs and memory. This is important because RPEs and mood state bidirectionally affect each other[20], and aberrant reward circuitry and altered RL are features of a range of mood and anxiety disorders[21,22]. These disorders can also result in impaired or biased memory processes[23,24]. While one prior study suggests that depression scores alter the relationship between RPE and memory[25], it is not clear whether this effect is specific to RPE-mediated memory, or which specific depressive symptoms might underlie this effect. These studies, in sum, raise the possibility that altered mood states might alter the RPEs-mediated memory. Therefore, understanding the computational mechanisms that link RPEs to memory could

[1]Department of Psychiatry, Icahn School of Medicine at Mount Sinai, New York, NY, USA. [2]Center for Computational Psychiatry, Icahn School of Medicine at Mount Sinai, New York, NY, USA. [3]The Winsor School, Boston, MA, USA. [4]Department of Neuroscience, Icahn School of Medicine at Mount Sinai, New York, NY, USA. [5]Department of Neurosurgery, Icahn School of Medicine at Mount Sinai, New York, NY, USA. [6]Department of Neurology, Icahn School of Medicine at Mount Sinai, New York, NY, USA. ✉e-mail: xiaosi.gu@mssm.edu

aid in assessing the biasing effect mood disorders have on memory, and developing therapeutic approaches to these symptoms and disorders.

## Methods

### Data collection and participants

The study was approved by the Institutional Review Board at the Icahn School of Medicine at Mount Sinai. Participants were recruited from Prolific (http://prolific.co), an online survey platform. A total of 246 adults (126 female, age = 40.1 ± 14.3 years) provided informed consent and completed this study. We excluded 40 participants whose overall behavioral performance in the decision-making task was no different from chance. The final sample that completed the behavioral task had 206 adults (109 female, age = 40.3 ± 14.2 years). Participants were asked to return to complete psychometric surveys. The sample that completed the psychometric surveys had 173 adults (94 female, age = 42.4 ± 14 years). Participants were paid a fixed rate with a bonus computed as a function of the reward accumulated in the first task in the experiment. The target sample size was determined based on the results of similar studies investigating the effect of RPE on memory[11,12]. The study was not pre-registered.

### Task

Participants performed an experiment with two distinct stages: a decision-making task, followed by a recognition memory task. The decision-making task was a two-arm bandit task in which participants attempted to maximize their rewards by learning one of two possible options (decks of cards) on each trial, for a total of 60 trials. Each draw could result in winning either 100 points or 0 points. Each option was associated with a specific win probability (either 0.8 or 0.2), which was reversed four times every 12 ± 1 trial. As a result, each option's win probability was always negatively correlated. However, in contrast to traditional bandit/reversal tasks, participants were shown a unique image stimulus after making each choice, associating each reward outcome with a specific image stimulus. These image stimuli were memorable faces drawn from a database on perceptual memorability (10k US Adult Faces database: https://wilmabainbridge.com/facememorability2.html)[26,27], where each face image was associated with a normed d' score[28] that measured how well these images were recognized in a large population sample. Face stimuli were chosen in part to be incidental to the decision-making task in order to maximize dissociation between stimulus features and decision-making behavior[12,29]; in contrast, participants were informed that they may need to remember the faces for a subsequent task. Participants were only paid for their performance in the decision-making task, however, to ensure there was no direct, instructed link between memory performance and reward attainment. Upon completing the decision-making task, participants immediately began a recognition memory task in which these 60 image stimuli were shown, in addition to 60 novel lure images drawn from the same memorability database with matched d' scores in random order. During this task, participants were instructed to indicate whether the image was "old" or "new", and then asked to indicate their confidence in their selection. We computed d', a signal-detection metric, for each subject by subtracting the z-score corresponding to the false-alarm rate from the z-score corresponding to the hit rate[28]. The task was constructed using the PsychoPy toolbox[30].

### Computational modeling

We utilized a Rescorla–Wagner model to fit behavior in the decision-making task, in which RPEs modulate a learning rate ($\alpha$) parameter, and RPE-based decisions are determined by an inverse-temperature parameter ($\beta$) modulating a softmax choice function. The learning and decision rules for this model are described by the following equations[31]:

$$Q_{t+1}^c = Q_t^c + \alpha(r_t - Q_t^c) \tag{1}$$

$$p_t^c = \frac{e^{\beta Q_t^c}}{\sum_{i=1}^C e^{\beta Q_t^c}} \tag{2}$$

**Table 1 | Model details**

| Model | Parameters |
|---|---|
| **RW** | α, β |
| WSLS | ε |
| Bayes | p(reward), p(switch) |

Models, along with free parameters, are utilized for model comparison. The bold row denotes the winning model.

where $Q_t^c$ is the value of the chosen option on trial t, updated according to $r_t$, the model-estimated continuous RPE values on every trial. In addition to the Rescorla–Wagner model, which caches values for trial-by-error decision-making, we also constructed alternative models to capture heuristic switching behavior and Bayesian estimation of task reward state. In the heuristic model, agents keep selecting a choice until they lose, at which point they shift to the other choice, with one free parameter ($\epsilon$) capturing choice bias (Table 1). The Bayesian filter model is based on two hidden states: one in which the purple deck is the correct choice, and the other in which the orange deck is the correct choice, with some probability that states have reversed on each trial. The model computes the likelihood that a choice is correct or incorrect as a function of the inferred probability of reward for the current state. Action probabilities are computed from this likelihood, taking into account the inferred probability that a state switch (e.g., a reward reversal) has occurred[32]. The free parameters for this model are the probability of reward, and the probability of reversal (Table 1).

To model the influences of RPE and perceptual memorability on memory search during recognition, we utilized drift-diffusion models (DDMs), which fit a noisy sequential sampling process to choice data such that relative evidence is accumulated over time until reaching a decision boundary (e.g., a recognition choice)[33]. We first excluded reaction times beyond 3 standard deviations away from the subject-level mean, and/or those shorter than 300 ms or exceeding 10 seconds. Then, in two separate hierarchical DDMs, we modeled drift rate (v), the rate of evidence accumulation prior to making a recognition choice, as a function of RPE or PM, with subject as a random effect:

$$v \sim \text{RPE} + (1|\text{subject}) + (\text{RPE}|\text{subject}) \tag{3}$$

$$v \sim \text{PM} + (1|\text{subject}) + (\text{PM}|\text{subject}) \tag{4}$$

The remaining free model parameters, including non-decision time (t), starting point (z), and boundary separation (a) were fit with complete pooling across participants. We utilized a hierarchical approach due to the relatively low number of trials contributed by each participant[34]. In addition, we also constructed alternative models to capture non-linear influences of RPE on drift rate, including polynomial and logarithmic relationships between RPE and drift rate:

$$v \sim \text{RPE}^2 + (1|\text{subject}) + (\text{RPE}^2|\text{subject}) \tag{5}$$

$$v \sim \log(\text{RPE}) + (1|\text{subject}) + (\log(\text{RPE})|\text{subject}) \tag{6}$$

### Bayesian mixed-effects regression

To determine the features predicting successful memory retrieval, we used a Bayesian mixed-effects logistic regression modeling framework[35]. Within this framework, we coded hits and correct rejections as correct memory choices and misses and false alarms as incorrect memory choices. We then constructed models of the form:

$$p(\text{correct} = 1) \sim X + (1|\text{subject}) + (\text{RPE}|\text{subject}) + (\text{PM}|\text{subject}) \tag{7}$$

where the probability of correct memory choices is modeled using a logit-link function of fixed effects (X) and random effects. The fixed effects

include the following trial-level predictors: RPE, PM, and the within-block trial number (coded such that trials following a reversal restart at 1). The fixed effect also includes subject-level traits, including age, sex, RL parameters ($\alpha$, $\beta$), total gambling reward, and factor scores. The random effects allow the influence of RPE and perceptual memorability to vary across subjects, as well as allow for a random intercept such that one intercept is fit per subject. All numerical predictors were standardized by subtracting the mean and dividing by two times the standard deviation[36], while sex was coded as a categorical variable. We generated weakly informative (broad) priors for all regression variables[37] which are scaled to regularize the model rather than integrate domain knowledge.

## Model fitting and assessment
Behavioral models and Bayesian mixed-effects regression models were fit to individual subject data using Bayesian inference over the free parameters, using the Python library pymc[38] and bambi[39]. To fit models, we used four Markov chain Monte Carlo No-U-Turn (NUTS) samplers, drawing 4000 samples from the posterior for each chain, after a minimum of 4000 burn-in samples. All posteriors for independent variables were checked for convergence using the Gelman–Rubin statistic, which was less than 1.01 in all cases, indicating good convergence. We computed the 95% high-density interval (HDI) for each model parameter to quantify the uncertainty around the true value of the parameter[40]. We considered there to be substantial evidence for the influence of a parameter if the 95% HDI did not include zero[41]. Model comparison was performed using the Waikake-information criterion[42]. When assessing parameter recoverability for the decision-making task, we used each model to simulate behavior for 206 agents utilizing the true parameters sampled from our cohort. Because the DDM model was hierarchical, we simulated 50 cohorts of 25 participants (a total of 1250 simulations) and fit each simulated cohort hierarchically. We fit this simulated behavior and computed the correlation between the original parameters used to simulate the behavior and those recovered by the fitting procedure. To test model identifiability, we used the fit parameters for each model to simulate behavior for 206 agents, fit this behavior using every model, and performed model comparison to determine which model fit the simulated behavior best.

## Factor analysis
We utilized factor analysis to identify latent transdiagnostic structure across three surveys: the state-trait anxiety index (STAI-T), the Zung depression scale (SDS), and the obsessive-compulsive inventory (OCI-R). These were selected to match factor analyses in prior literature[43]. We first needed to increase the sample size to ensure a robust estimation of factor loadings and scores. To do so, we utilized survey data from an additional 143 online participants who had completed the same set of surveys as our task participants, bringing the total number of participants utilized for the factor analysis, specifically, to n = 320. First, we computed the Kaiser–Meyer–Olkin (KMO) measure of sampling adequacy to assess whether it was plausible to conduct a factor analysis and found that the degree of overlapping information among the survey responses was appropriate for a factor analysis (KMO = 0.94). We also computed Bartlett's sphericity test and found that the correlation matrix of the survey responses was not an identity matrix, and thus appropriate for a factor analysis ($\chi^2 = 13034$). We next performed factor analysis using an oblique promax rotation, using maximum likelihood estimation. We used the Cattell–Nelson–Gorsuch (CNG) test[44] to determine the appropriate number of factors for this data, verified by the resulting scree plot showing the first three factors captured the most variance in eigenvalues (Supplementary Fig. 6). The factor loadings for each survey question are depicted in Table 2. The items with high factor loadings were used to categorize the factors into the following categories: positive affect, intrusive thoughts and rumination, and obsessive–compulsive behavior. We computed the factor score using the ten Berge method[45].

## Statistical analysis and software
Statistical analysis was conducted in Python, using publicly available libraries. Bayesian model-fitting was conducted using pymc, a Python library for Bayesian inference. Drift-diffusion modeling, specifically, was conducted using hssm, a Python library built on top of pymc for constructing sequential sampling models[34]. Bayesian mixed-effects modeling was conducted using bambi, a Python library built on top of pymc for constructing Bayesian regression models. Data distribution was assumed to be normal when using parametric statistical tests, but this was not formally tested. All null regression findings are accompanied by equivalence tests with equivalence bounds of [−0.1, 0.1].

## Reporting summary
Further information on research design is available in the Nature Portfolio Reporting Summary linked to this article.

# Results
## Behavioral analysis and computational modeling of decision-making behavior
In order to investigate how the RPEs that become associated with a stimuli influence our memories for that stimuli, we designed an experiment consisting of two consecutive tasks: a decision-making task followed by a memory task, similar to prior studies[9–13]. For the decision-making task, participants performed a two-arm bandit task (Fig. 1a), in which they drew cards from two separate decks with oppositely yoked reward probabilities that reversed four times throughout the task without warning (80% chance of reward for one deck, 20% chance of reward for the other)[46]. After every decision, participants were shown reward feedback (either 0 or 100 points) for their choice, along with unique image stimuli. These image stimuli were then utilized in the subsequent memory task, where participants were asked to indicate if a cue image had been seen before, or was a novel lure. We selected the image stimuli from a database of face images with normed perceptual memorability ratings (PM)[26] (see "Methods"). Memorability is an intrinsic, perceptual property of images that is predictive of how easily remembered an image is[27], and thought to potentially reflect how perceptual information is prioritized for memory[47]. As such, the memorability ratings of these images provided a metric for measuring how successful memory might also fluctuate as a function of the perceptual information associated with each image, independent of the extrinsic RPE that participants encoded along with each image.

We recruited 246 online participants (126 female) from Prolific (http://prolific.com), an online survey platform, to perform this experiment. After excluding participants with below chance-level accuracy for the decision task, we demonstrated that the remaining participants ($n = 206$) learned to choose the more rewarding option during the decision-making task, even after reward probabilities reversed (Fig. 1b). These participants subsequently completed the recognition memory task (Fig. 1c), in which they have to assess whether stimuli in a set had been presented before, or are novel. Accuracy and reaction times (RTs) in the two tasks were correlated ($\rho = 0.16$, $p = 0.02$, $\rho = 0.49$, $p < 0.001$, respectively; Supplementary Fig. 1a, b), though RTs for the decision-making task were significantly faster than during the memory task ($t(410) = -29.5$, $p < 0.001$, Cohen's $d = 2.9$, 95% CI = [−1.1, −0.9]). In addition to successful recognition choices, called hits (Fig. 1d), memory responses were additionally categorized as correct rejections, misses, or false alarms (Supplementary Fig. 1C). Memory performance computed from a combination of these response categories ($d'$) indicated that participants performed above chance in the memory task (Supplementary Fig. 1D), though memory performance tended to decrease towards the end of the recognition period and was asymmetric between old and novel lure images ($z = 31.8$, $p < 0.001$, Cohen's $h = 1.9$, 95% = [29.5, 34]; Supplementary Fig. 1a, b).

Having demonstrated that participants exhibited sufficient learning in the first task and memory in the second (Supplementary Fig. 1D),

**Table 2 | Transdiagnostic factors**

| Factor 1: positive affect | loading | Factor 2: intrusive thoughts and rumination | loading | Factor 3: obsessive-compulsive behavior | Loading |
|---|---|---|---|---|---|
| I am happy | 0.90 | I get in a state of tension or turmoil as I think over my recent concerns and interests | 0.84 | I feel compelled to count while I am doing things | 0.72 |
| I feel pleasant | 0.86 | Some unimportant thoughts run through my mind and bother me | 0.80 | I repeatedly check doors, windows, drawers, etc. | 0.71 |
| I am content | 0.85 | I take disappointments so keenly that I can not put them out of my mind | 0.75 | I need things to be arranged in a particular way | 0.69 |
| I am calm, cool, and collected | 0.79 | I feel nervous and restless | 0.73 | I feel I have to repeat certain numbers | 0.66 |
| I feel that I am useful and needed | 0.77 | I am upset by unpleasant thoughts that come into my mind against my will | 0.65 | I wash my hands more often and longer than necessary | 0.64 |
| I feel rested | 0.76 | I have disturbing thoughts | 0.65 | I repeatedly check gas and water taps and light switches after turning them off | 0.64 |
| I feel secure | 0.74 | I find it difficult to control my own thoughts | 0.64 | I sometimes have to wash or clean myself simply because I feel contaminated | 0.60 |
| I still enjoy the things I used to do | 0.73 | I worry too much over something that does not really matter | 0.64 | I get upset if objects are not arranged properly | 0.60 |
| I feel satisfied with myself | 0.73 | I frequently get nasty thoughts and have difficulty in getting rid of them | 0.61 | I find it difficult to touch an object when I know it has been touched by strangers or certain people | 0.59 |
| I am a steady person | 0.73 | I feel downhearted and blue | 0.60 | I get upset if others change the way I have arranged things | 0.58 |
| I feel hopeful about the future | 0.68 | I feel that difficulties are piling up so that I cannot overcome them | 0.59 | I check things more often than necessary | 0.49 |
| My life is pretty full | 0.68 | I have saved up so many things that they get in the way | 0.55 | I collect things I do not need | 0.44 |
| I make decisions easily | 0.67 | I feel like a failure | 0.55 | I feel that there are good and bad numbers | 0.43 |
| I find it easy to make decisions | 0.66 | I feel inadequate | 0.54 | I avoid throwing things away because I am afraid I might need them later | 0.37 |
| I still enjoy sex | 0.62 | I have crying spells or feel like it | 0.54 | I am upset by unpleasant thoughts that come into my mind against my will | 0.26 |
| I find it easy to do the things I used to | 0.62 | I wish I could be as happy as others seem to be | 0.54 | I frequently get nasty thoughts and have difficulty in getting rid of them | 0.24 |
| My mind is as clear as it used to be | 0.58 | I lack self-confidence | 0.53 | I find it difficult to control my own thoughts | 0.24 |
| I eat as much as I used to | 0.41 | My heart beats faster than usual | 0.49 | I have saved up so many things that they get in the way | 0.19 |
| Morning is when I feel the best | 0.30 | I feel that others would be better off if I were dead | 0.48 | I feel satisfied with myself | 0.18 |
| Some unimportant thoughts run through my mind and bother me | 0.11 | I am more irritable than usual | 0.45 | I have trouble sleeping at night | 0.13 |
| My heart beats faster than usual | 0.08 | Morning is when I feel the best | 0.43 | Morning is when I feel the best | 0.12 |
| I get tired for no reason | 0.07 | I am restless and can not keep still | 0.40 | I am happy | 0.12 |
| I collect things I do not need | 0.07 | I get tired for no reason | 0.28 | I get tired for no reason | 0.12 |
| I am upset by unpleasant thoughts that come into my mind against my will | 0.07 | I collect things I do not need | 0.27 | My heart beats faster than usual | 0.09 |
| I frequently get nasty thoughts and have difficulty in getting rid of them | 0.07 | I avoid throwing things away because I am afraid I might need them later | 0.27 | I notice that I am losing weight | 0.08 |
| I have saved up so many things that they get in the way | 0.06 | I notice that I am losing weight | 0.06 | I feel secure | 0.07 |
| I feel that others would be better off if I were dead | 0.06 | I have trouble with constipation | 0.06 | I am content | 0.06 |
| I notice that I am losing weight | 0.04 | I still enjoy sex | 0.06 | I feel pleasant | 0.06 |
| I have disturbing thoughts | 0.04 | I have trouble sleeping at night | 0.04 | My life is pretty full | 0.06 |
| I feel that there are good and bad numbers | 0.04 | I check things more often than necessary | 0.04 | I am restless and can not keep still | 0.06 |
| | | I eat as much as I used to | 0.04 | | |

## Table 2 (continued) | Transdiagnostic factors

| Factor 1: positive affect | loading | Factor 2: intrusive thoughts and rumination | loading | Factor 3: obsessive-compulsive behavior | Loading |
|---|---|---|---|---|---|
| I feel compelled to count while I am doing things | 0.04 | I feel that there are good and bad numbers | 0.09 | I am calm, cool, and collected | 0.06 |
| I sometimes have to wash or clean myself simply because I feel contaminated | 0.04 | I find it easy to do the things I used to | 0.06 | I feel hopeful about the future | 0.02 |
| I find it difficult to control my own thoughts | 0.02 | I get upset if others change the way I have arranged things | 0.05 | I am a steady person | 0.02 |
| I get in a state of tension or turmoil as I think over my recent concerns and interests | 0.01 | I sometimes have to wash or clean myself simply because I feel contaminated | 0.04 | I feel that I am useful and needed | 0.02 |
| I wash my hands more often and longer than necessary | 0.00 | I find it difficult to touch an object when I know it has been touched by strangers or certain people | 0.03 | I feel rested | 0.02 |
| I find it difficult to touch an object when I know it has been touched by strangers or certain people | 0.00 | I get upset if objects are not arranged properly | 0.03 | Some unimportant thoughts run through my mind and bother me | 0.01 |
| I need things to be arranged in a particular way | -0.02 | I am happy | 0.03 | I am more irritable than usual | 0.01 |
| I get upset if others change the way I have arranged things | -0.03 | I feel pleasant | 0.03 | I have crying spells or feel like it | 0.00 |
| I repeatedly check doors, windows, drawers, etc. | -0.04 | I make decisions easily | 0.01 | I have trouble with constipation | 0.00 |
| I get upset if objects are not arranged properly | -0.04 | My mind is as clear as it used to be | -0.01 | I feel that others would be better off if I were dead | -0.01 |
| I avoid throwing things away because I am afraid I might need them later | -0.04 | I feel rested | -0.01 | I have disturbing thoughts | -0.02 |
| I am restless and can not keep still | -0.04 | I repeatedly check gas and water taps and light switches after turning them off | -0.02 | I make decisions easily | -0.02 |
| I check things more often than necessary | -0.05 | I feel I have to repeat certain numbers | -0.02 | I get in a state of tension or turmoil as I think over my recent concerns and interests | -0.04 |
| I have crying spells or feel like it | -0.05 | I still enjoy the things I used to do | -0.02 | I worry too much over something that does not really matter | -0.04 |
| I repeatedly check gas and water taps and light switches after turning them off | -0.06 | I am content | -0.02 | I take disappointments so keenly that I can not put them out of my mind | -0.06 |
| I take disappointments so keenly that I can not put them out of my mind | -0.07 | I need things to be arranged in a particular way | -0.04 | I find it easy to make decisions | -0.08 |
| I worry too much over something that does not really matter | -0.09 | I am calm, cool, and collected | -0.05 | I feel that difficulties are piling up so that I cannot overcome them | -0.08 |
| I feel nervous and restless | -0.13 | I find it easy to make decisions | -0.05 | I feel downhearted and blue | -0.08 |
| I have trouble with constipation | -0.16 | I feel hopeful about the future | -0.05 | I still enjoy the things I used to do | -0.09 |
| I get tired for no reason | -0.18 | I wash my hands more often and longer than necessary | -0.05 | I wish I could be as happy as others seem to be | -0.10 |
| I am more irritable than usual | -0.22 | Morning is when I feel the best | -0.10 | I feel nervous and restless | -0.10 |
| I have trouble sleeping at night | -0.22 | I feel that I am useful and needed | -0.10 | I lack self-confidence | -0.11 |
| I feel downhearted and blue | -0.25 | I am a steady person | -0.11 | I still enjoy sex | -0.13 |
| I wish I could be as happy as others seem to be | -0.25 | I feel compelled to count while I am doing things | -0.12 | I feel like a failure | -0.14 |
| I lack self-confidence | -0.28 | My life is pretty full | -0.13 | My mind is as clear as it used to be | -0.18 |
| I feel like a failure | -0.35 | I repeatedly check doors, windows, drawers, etc. | -0.14 | I feel inadequate | -0.20 |
| I feel inadequate | -0.36 | I feel secure | -0.16 | I eat as much as I used to | -0.20 |
|  |  | I feel satisfied with myself | -0.18 | I find it easy to do the things I used to | -0.23 |

Factor loadings, sorted by factor, for each survey question.

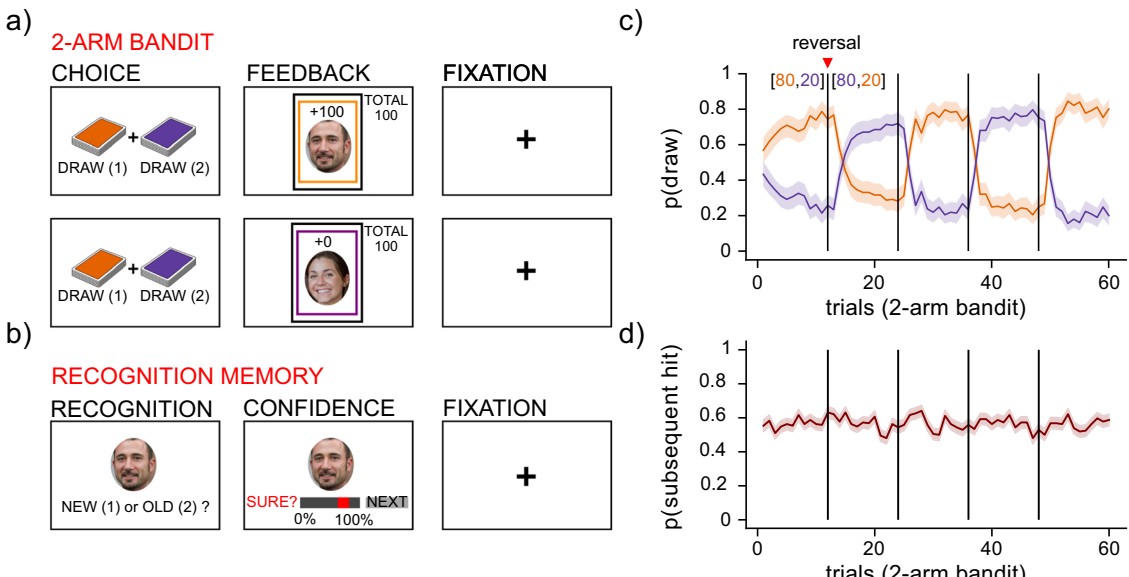

**Fig. 1 | Task schematic and participant performance. a** Schematic of two representative trials of the two-arm bandit task. Participants were cued to select a card from one of two decks (orange or purple) on each trial, then provided with feedback and a trial-unique face image, followed by a fixation cross. **b** Probability of drawing from the male (orange) or female (purple) decks as a function of trial and reward block (mean reversal trial indicated by vertical lines). Shaded lines denote a 95% confidence interval. **c** Schematic of the representative trial of the recognition memory task. Participants were presented with images of faces, and instructed to decide whether the images were old or novel lures. Participants then had to indicate their confidence in their selection, followed by a fixation cross. **d** Probability of a successful subsequent recognition (hit) for each image seen during the decision-making task, as a function of decision-making trial (mean reversal trial indicated by vertical lines). Shaded lines denote a 95% confidence interval.

we constructed computational models of participants' choices during the decision-making task to investigate how RPEs might impact their learning and, subsequently, their memory. Learning and decision-making in similar tasks are well captured by reinforcement learning models driven by RPEs[48,49]. As such, we utilized a Rescorla-Wagner model, an RL model driven by trial-and-error learning from incoming RPEs, to fit behavior in the decision-making task (see "Methods" for model details). In addition, we tested this model against alternative models that do not rely on cached value or RPEs, including a heuristic win-stay, lose-shift model, and a Bayesian filter model estimating the probability of reward for correct choices as well as the probability of reward reversal[46,50] (Table 1). We performed a model comparison (see "Methods") to select the model that provided the best and most parsimonious fit for the majority of participants' data. The winning model was the Rescorla-Wagner (RW) model ($\chi^2 = 18.5$, $p < 0.001$, Cramer's $v = 0.02$, chi-square test of proportions, Supplementary Fig. 3A–C) with two free parameters: a learning rate, $\alpha$, which dictates how strongly RPEs influence value assignment, and inverse temperature, $\beta$, which dictates how deterministically value assignments influence choice.

The best-fit parameters across the sample included a learning rate of $0.73 \pm 0.19$, suggesting that RPEs were weighed heavily in estimating the value of each option and an inverse temperature of $4.9 \pm 2.9$ (Fig. 2a). We performed a grid search over the joint parameter space, simulating actions, and outcomes. The combination of parameters that maximized reward in these simulations was a combination of a high learning rate and high inverse temperature[51]. Accordingly, a comparison between the subset of participants with the highest combination of these parameters vs. the lowest combination (categorized by quantile split) illustrated a dissociation in optimal behavior during the first and second half of each block (Fig. 2b), resulting in higher reward for the participants with the higher combination of learning rate and inverse temperature after learning had stabilized within each block ($z = 2.0$, $p = 0.047$, Cohen's $h = 1.9$, 95% CI = [−0.26, 4.2]; Fig. 2c).

## Reward-prediction errors and perceptual information separably enhance recognition memory

Next, we investigated how the RPE and perceptual memorability ratings associated with each stimulus affected the memory of the stimulus on a trial-by-trial basis. Previous studies that have focused on the trial-level enhancement of memory for stimuli focused on either the contribution of intrinsic perceptual information[26] or extrinsic reward information[9-13,29]. In contrast, our task design and stimuli choice enabled us to associate each stimulus with both a model-estimated RPE as well as a rating based on each stimulus' normed intrinsic memorability in the absence of rewards[26] (Fig. 2c). We selected only stimuli considered highly memorable on average (those with high perceptual memorability ratings, see "Methods") to ensure that participants could achieve high recognition success even if ignoring RPEs entirely. We first confirmed that perceptual memorability ratings and model-estimated RPEs were orthogonal ($\rho = -0.02$, Fig. 2d) and plotted their joint contribution to the probability of correct recognition (Fig. 2e) to visualize the relative contribution of both streams of information. The group-level and subject-level relationships between RPE and memory and perceptual memorability and memory replicated prior studies investigating the effects of these individual features on hit probability[9,10,12,13] (Supplementary Fig. 4). We next sought to understand the parallel contributions of RPE and perceptual memorability to memory beyond the probability of hits alone, while simultaneously accounting for subject-level RL parameters, demographics, and random-effects. To this end, we utilized a Bayesian mixed-effects logistic regression model to measure the importance of extrinsic RPE information and intrinsic perceptual information to correct vs. incorrect memory performance accounting for all four types of memory responses (hits, correct rejections, misses, and false alarms; see "Methods").

Across participants, the RPE and perceptual memorability associated with each stimulus meaningfully contributed to memory (fixed effects posterior mean = 0.17, 0.16, 95% HDI = [0.11, 0.23], [0.07, 0.25], respectively; Fig. 3a), such that surprisingly rewarding stimuli and more perceptually memorably stimuli were remembered better than other stimuli.

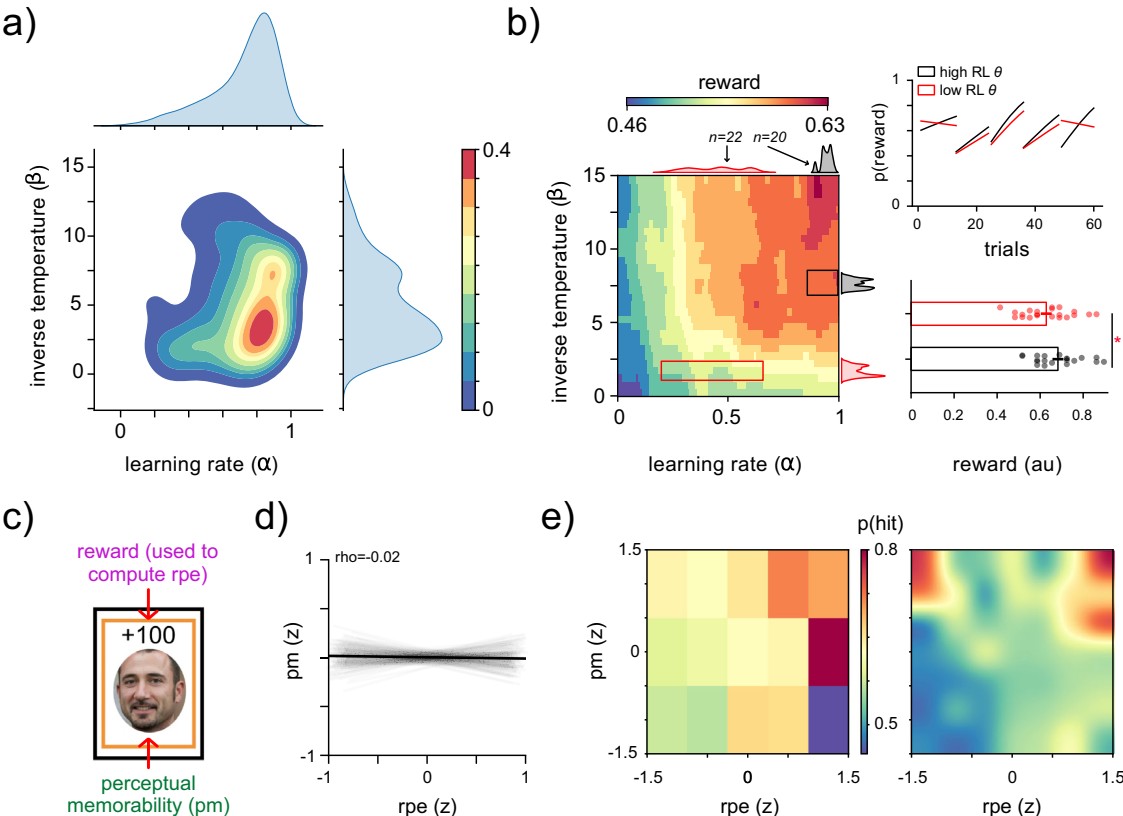

**Fig. 2 | Model parameters. a** Joint and marginal distributions of parameter estimates from the RW model, depicting the learning rate ($\alpha$) and inverse temperature ($\beta$). Warm colors indicate higher density. **b** Left: simulated reward outcomes for different combinations of parameters. Warm colors denote better performance (more reward). Inset boxes indicate subset of participants with larger parameters (black, $n = 20$), and smaller parameters (red, $n = 22$). Marginal distributions of parameters are indicated for each subset. Right, top: logistic fit of reward outcome as a function of trial within each block (prior to reversal), for participants with larger parameters (black) vs smaller parameters (red). Right bottom: Comparison of total reward accumulated for participants during the second half of each block with larger (black) vs smaller (red) parameters, Asterisk indicates significant difference ($z = 2.0$, $p = 0.047$, Cohen's $h = 1.9$, 95% CI = [$-0.26$, 4.2]). Dots indicate mean performance for individual subjects. **c** Schematic of stimulus features contributing to subsequent memory. Each stimulus shown during the decision-making task was associated with both a model-estimated RPE (purple) and normed perceptual memorability rating (green). **d** Linear fit of the relationship between RPE and perceptual memorability ratings for each stimulus. Shaded lines denote fit for individual participants, while solid line indicates fit across participants ($B = -0.01$, SE = 0.01, $p = 0.13$, $t_{eq}(12358) = 12.5$, $p_{eq} < 0.001$, 95% CI = [$-0.03$, 0.004]). **e** Binned (left) and smoothed (right) probability of a correct recognition (hit) across all trials and participants, as a function of both image RPE and image PM. Warm colors denote better performance (more hits).

Furthermore, stimuli that appeared sooner after the reward-probability reversal were remembered better than those that appeared later after the reversal (fixed effects posterior mean = $-0.12$, 95% HDI = [$-0.19$, $-0.01$]), suggesting that proximity to state changes also induced increased memorability. Participant demographics and RL parameters (learning rate and inverse temperature) did not meaningfully predict memory performance in this full model, nor did the total reward earned during the decision-making task. To examine how RPE and perceptual memorability influence a different index of memory behavior, we utilized drift-diffusion models (DDM) fit to participants reaction time and choices during the recognition memory task (Fig. 3b, Supplementary Fig. 5A). Specifically, we assessed whether RPE or perceptual memorability more strongly modulated drift rate (Fig. 3B, Supplementary Fig. 5b)—if either RPE or perceptual memorability upregulated drift rate, it would suggest that this feature contributes positively to evidence accumulation in support of the recognition of the target image in opposition to evidence accumulating against it. The model integrating RPE was preferred to the model integrating perceptual memorability (Fig. 3b; see Supplementary Fig. 5C, D for all posterior estimates and parameter recovery). This suggests that RPE explained more variance in participant recognition responses and reaction time than PM, though both RPE (posterior mean = 0.046, 95% HDI = [0.025, 0.067]) and perceptual memorability (posterior mean = 0.038, 95% HDI = [0.021, 0.056]) contributed positively to drift rate (Fig. 3c). We computed several alternative models

testing non-linear (e.g., logarithmic and polynomial) relationships between RPE, perceptual memorability and drift-rate; however the linear model fit the behavioral data best (Supplementary Fig. 5E).

**Positive affect upregulates the beneficial effects of reward-prediction error on memory**

Next, we were interested in determining whether individual affective phenotype modulated the link between reward prediction, perceptual information, and memory to ascertain the features that might contribute to altered memory processes in mood disorders. We thus next examined whether individual differences in self-reported affective symptoms might modulate subject-level reliance on perceptual and reward information during memory. Specifically, we collected the following psychometric surveys from a subset of task participants ($n = 173$): the state-trait anxiety index (STAI-T), the Zung depression scale (SDS), and the obsessive-compulsive inventory (OCI-R) (Fig. 4A). Similar to prior studies[43,52], we utilized factor analysis to identify latent, transdiagnostic constructs and to derive synthesized affective symptom scores given the considerable overlap between depression and anxiety symptoms and effects on cognition (see "Methods"). Factor analysis identified three prominent factors (Supplementary Figs. 6 and 7A): positive affect (factor 1; e.g., "I am content"), intrusive thoughts and rumination (factor 2; e.g., "I lack self-confidence"), and obsessive-compulsive behaviors (factor 3; e.g., "I feel I have to repeat certain

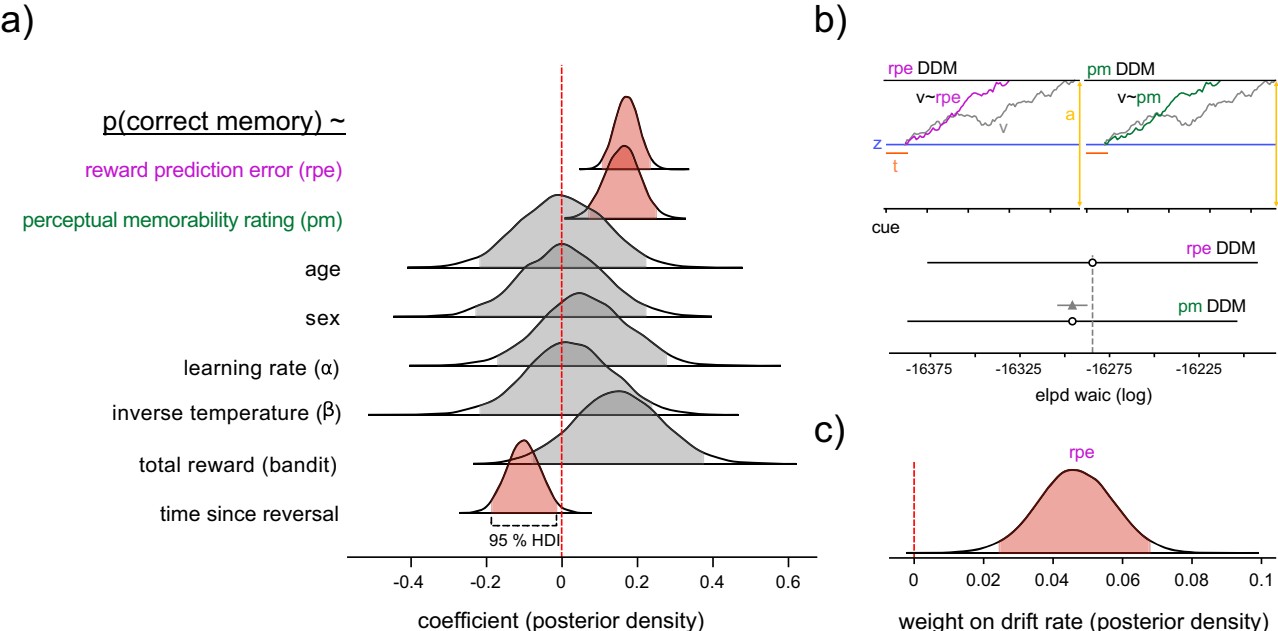

**Fig. 3 | Reward prediction errors and perceptual information make separable contributions to memory. a** Posterior distributions for fixed effects in a mixed-effects model examining how different factors influence recognition memory. The shaded portion represents the 95% high-density (HDI) interval. The vertical line indicates a coefficient of 0. Posterior distributions that include 0 are shaded gray, while those that do not are shaded red, indicating a meaningful effect (fixed effects posterior mean = 0.17, 0.16, −0.12, 95% HDI = [0.11, 0.23], [0.07, 0.25], [−0.19,

−0.01], respectively). **b** Top: Schematic of drift-diffusion models of recognition reaction time and choice that allow RPE (purple) or perceptual memorability (green) to influence the drift rate (v). Bottom: Model comparison of the RPE and perceptual memorability DDM models, showing that the model using RPE fits the empirical data best. The gray triangle indicates the difference in WAIC scores. **c** Posterior distribution for the weight indicating the influence of RPE on drift rate, shaded red to indicate a meaningful effect (posterior mean = 0.046, 95% HDI = [0.025, 0.067]).

numbers") (Fig. 4B, Table 2). Factor loadings for the factors were allowed to correlate ($\rho_{F1:F2}$ = −0.62, $p < 0.001$, $\rho_{F1:F3}$ = −0.22, $p = 0.1$, $\rho_{F2:F3}$ = −0.38, $p = 0.003$). We next sought to understand how subject-level factor scores (f1, f2, f3) predicted trial-level memory performance. Because these three factors were correlated, we utilized three separate Bayesian mixed-effects models and performed model comparisons to identify which of these factors explained the most variance in memory performance (see "Methods"). The model including factor 1 (positive affect) performed marginally better than the models including factor 2 (intrusive thoughts and rumination) or factor 3 (obsessive–compulsive behaviors; Fig. 5a). Subjects' factor 1 score did not modulate trial-level memory performance, or interact with trial-level RPE (main effect 95% HDI include 0; Fig. 5a). Because the model including only factor 1 performed best, we used the regression coefficients from this mixed-effects model for subsequent analyses.

While the mixed-effects model explored how factor scores influenced trial-level memory, we next examined whether subject-level individual differences in memory were explained by individual differences in factor scores and subject-level reliance on RPE for memory ($\beta_{RPE}$; the slope of the relationship between RPE and memory estimated for each subject in the mixed-effects model). Following the trial-level results, we thus performed a regression analysis of subject-level memory performance as a function of participants' reliance on RPE for memory ($\beta_{RPE}$) and continuous factor scores and found that subjects' factor 1 score exhibited a significant interaction with subjects' overall reliance on RPE for memory ($B = 1.4$, SE = 0.5, $p = 0.01$, 95% CI = [0.3, 2.5]). This subject-level result indicated that participants who relied more on RPEs for memory also exhibited better memory overall—if they were also individuals with greater positive affect (higher f1 scores). We did not detect a significant interaction between positive affect and participants' reliance on perceptual memorability, but could not reject the presence of small effects based on an equivalence test ($B = −4.5$, SE = 4.5, $p = 0.32$, $t_{eq}(166) = 1.03$, $p_{eq} = 0.15$, 95% CI = [−13.5, 4.4]; Fig. 5b). Similarly, we did not detect significant relationships between positive affect and reward ($B = −0.02$, SE = 0.29, $p = 0.95$, $t_{eq}(168) = 0.41$, $p_{eq} = 0.34$, 95%

CI = [−0.6, 0.6]), learning rate ($B = −0.001$, SE = 0.01, $p = 0.94$, $t_{eq}(168) = 7$, $p_{eq} < 0.001$, 95% CI = [−0.03, 0.03], or inverse temperature ($B = −0.13$, SE = 0.21, $p = 0.52$, $t_{eq}(168) = 1.1$, $p_{eq} = 0.13$, 95% CI = [−0.6, 0.3]; Supplementary Fig. 7B), though we could not reject the presence of small effects based on equivalence tests. However, equivalence tests did confirm the absence of a significant relationship between positive affect and memory ($B = 0.05$, SE = 0.04, $p = 0.29$, $t_{eq}(168) = 7.2$, $p_{eq} < 0.001$, 95% CI = [−0.04, 0.13]), as well as mnemonic reaction time ($B = −0.02$, SE = 0.02, $p = 0.29$, $t_{eq}(168) = 6$, $p_{eq} < 0.001$, 95% CI = [−0.06, 0.02]; Supplementary Fig. 7B), suggesting that positive affect did not alter the relationship between RPE and memory by modifying overall memory performance or speed. We did observe that subjects' factor 2 score exhibited a significant interaction with subjects' overall reliance on perceptual memorability for memory ($B = 12.8$, SE = 5.2, $p = 0.015$, 95% CI = [2.5, 23]), suggesting that participants who relied more on perceptual information for memory exhibited better memory overall—if they were more anxious and ruminative (higher f2 scores). We did not observe a significant relationship between RPE reliance and memory for f2 ($B = −0.52$, SE = 0.56, $p = 0.36$, $t_{eq}(166) = 1.1$, $p_{eq} = 0.14$, 95% CI = [−1.6, 0.6] or f3 ($B = −0.3$, SE = 0.6, $p = 0.6$, $t_{eq}(166) = 0.7$, $p_{eq} = 0.24$, 95% CI = [−1.5, 0.9]; Supplementary Fig. 8A, B), but could not reject the presence of a small effect based on equivalence tests.

## Discussion

Identifying the specific relevance of RPEs to memory is crucial for understanding why we remember rewarding events better than others and how this process can go awry in psychiatric states featuring altered mood. To that end, we designed and tested an experimental paradigm in which participants performed a decision-making task followed by a memory task. Using a reinforcement learning model, we found that participants remembered a stimulus better if it was associated with a model-estimated RPE, or if it was a more perceptually memorable stimulus. By disentangling the relationship between RPE-driven memory and perceptually-driven memory, we were

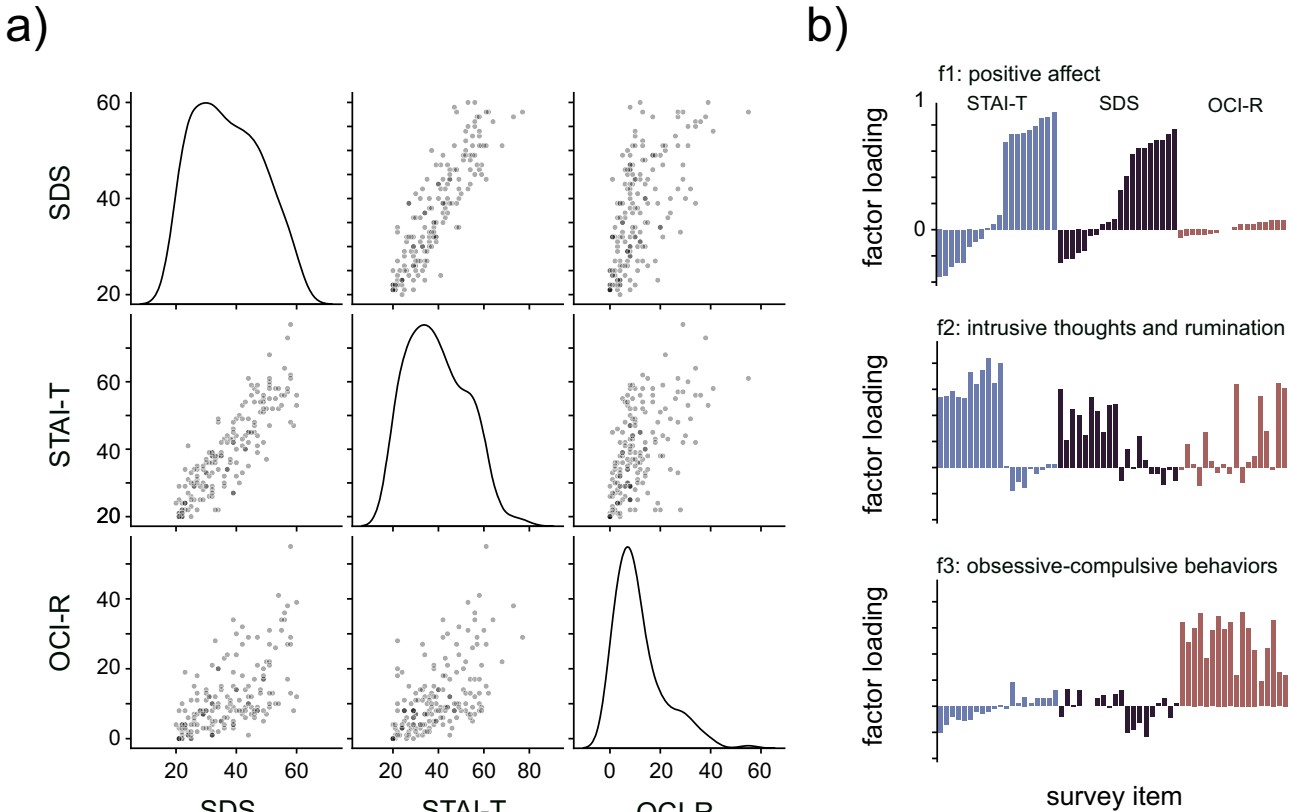

**Fig. 4 | Factor analysis reveals latent transdiagnostic constructs for mood.**
**a** Psychometric survey scores across participants. Diagonal: distribution of scores for depression (SDS scale), trait anxiety (STAI-T), and OCD (OCI-R) across participants. Off-diagonal: pairwise scatterplots depicting the relationship between scores across scales. **b** Results of factor analysis indicating factor loadings across survey questions for STAI-T (blue), SDS (black), and OCI-R (red) scales. Bars are ordered with respect to survey and ascending Factor 1 loading. Factor 1 (top) consists of questions primarily from the anxiety and depression scales related to positive affect. Factor 2 (middle) consists of questions across all three scales related to intrusive thoughts and rumination. Factor 3 (bottom) consists of questions primarily from the OCD scale related to obsessive behaviors.

able to observe differences in their behavioral consequences. Specifically, we showed that RPE improves the efficiency of successful recognition over perceptual information and that transdiagnostic measures of positive affect regulated the relationship between RPE-driven memory and enhanced memory performance, which was not true for perceptually-driven memory. Furthermore, this regulation was specific to affective phenotype and was not present using alternative mental health factors such as intrusive thoughts and rumination or obsessive-compulsive traits. Together, these findings illuminate the computational mechanisms mediating the important relationship between decision-making, memory, and affect.

While several prior studies have investigated whether model-estimated RPEs influence subsequent memories[9–13,53], these studies provide conflicting evidence for whether positive, negative, or unsigned RPEs enhance memory, whether this effect is most pronounced only after a delay period, or even whether these effects are age-dependent. One possible reason for these conflicting accounts is that prior studies do not account for how inherently memorable stimuli are on the basis of reward-agnostic perceptual features. The current task design specifically utilized stimuli with known perceptual memorability ratings[27] that index how intrinsically memorable a stimulus is in the absence of rewards. These memorability ratings are not explained by low-level visual properties, esthetic attractiveness, or interest level[19,26,27,54]. Furthermore, perceptual memorability ratings are not explained by purely attentional processes[15,27] and correlate with memorability scores determined by neural networks[47], suggesting that perceptual memorability captures intrinsic stimulus properties at the junction between high-level visual processing and memory. This information, in combination with computationally estimated RPEs, enabled us to dissociate between RPE-driven and

perceptually-driven memory processes in the subsequent recognition memory task using these same stimuli. While mixed-effects modeling demonstrated that both positive RPE and high perceptual memorability contributed to successful memory, drift-diffusion modeling of reaction times during memory revealed that positive RPEs more meaningfully up-regulated drift rate during memory search. While the strength of visual information is implicated in evidence accumulation in perceptual decision-making[55], our results suggest that the reward computations associated with surprising rewards provide more important evidence per unit of time for matching the recognition cue stimulus to the image stored in memory. This finding provides support for a functional dissociation between RPE- and perceptually-mediated memory enhancement.

Because perception and subjective valuation involve distinct neural circuits and cognitive processes, understanding their contributions to memory has distinct implications for the downstream effects that psychiatric disorders impairing perception or learning might have on memory. By showing that the transdiagnostic affective state modulates the link between reward information and memory but not perceptual information, our findings provide evidence that RPEs may bring a degree of salience to a stimulus that strengthens memory encoding. Identifying the specific influence that RPEs have on memory is critical to understanding the neural mechanisms that may underlie memory enhancement. In addition to providing computationally parsimonious models of decision making, reinforcement learning algorithms play a critical role in biological psychology because RPEs have been tightly correlated with the activity of dopaminergic neurons[3] in the substantia nigra and ventral tegmental area (VTA). This suggests that these models capture dopamine-driven learning[1,4]

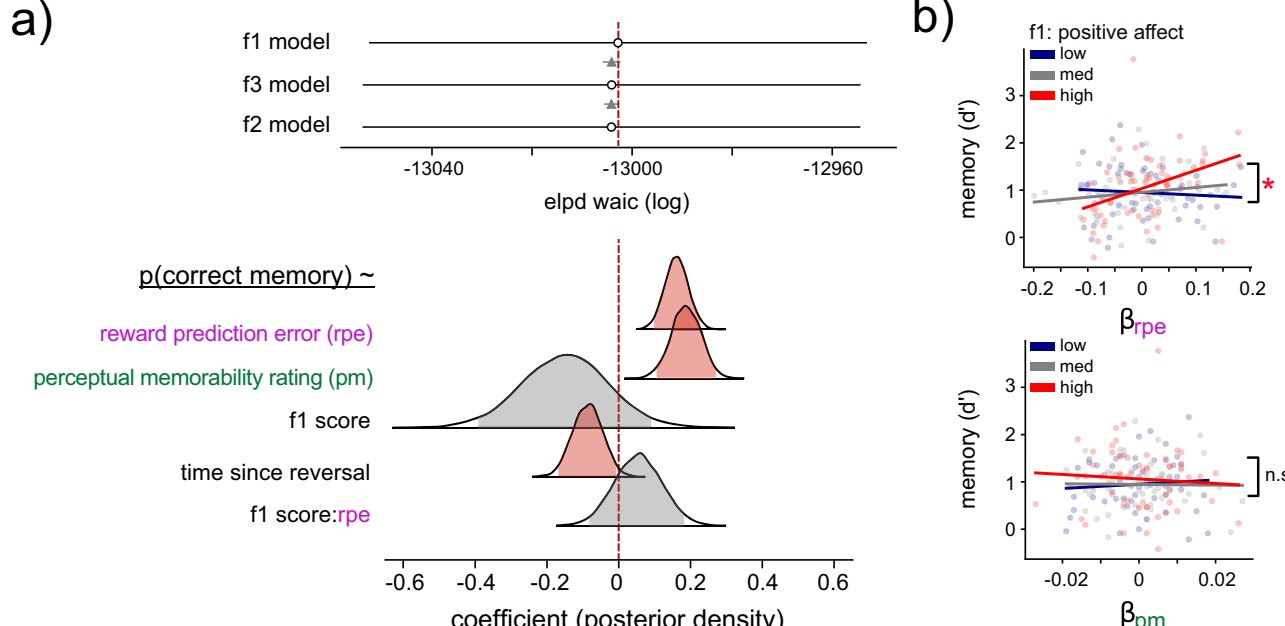

**Fig. 5 | Mood regulates the memory-enhancing effect of RPEs. a** Top: Model comparison of the three mixed-effects regression models showing that the model assessing the influence of factor 1 on memory fits the empirical data better than the model assessing the influence of factors 2 or 3. The gray triangle indicates the difference in WAIC scores. Bottom: Posterior distributions for fixed effects in mixed-effects model examining how transdiagnostic factor score f1 impacts memory. The shaded portion represents the 95% high-density (HDI) interval. The vertical line indicates a coefficient of 0. Posterior distributions that include 0 are shaded gray, while those that do not are shaded red, indicating a meaningful effect. **b)** Top: the relationship between $\beta_{RPE}$ (subject-level random effect from the mixed-effects model) and memory performance, organized by a tercile split (for visualization only) of positive affect (measured by factor 1 score). The asterisk indicates a significant interaction between factor score and $\beta_{RPE}$ in predicting memory performance (linear-regression coefficient = 1.4, $p = 0.011$). Bottom: the relationship between $\beta_{PM}$ (subject-level random effect from the mixed-effects model) and memory performance, arranged by a tercile split of positive affect (measured by factor 1 score). Dots denote values for individual participants. The solid line indicates linear model fit to participant data.

facilitated by mesocortical and cortico-striatal circuits for habits, action–selection, and decision-making. One prominent theory of RPE-mediated memory is that midbrain dopaminergic neurons innervate the hippocampus, and that dopamine release strengthens hippocampal plasticity involved in memory encoding and consolidation[5]. In support of this theory, neurophysiological evidence from rodents has demonstrated that mesolimbic dopamine modulates memory-related neuronal activity during memory encoding[56,57]. Research has also shown that neuronal activity in the VTA, a critical region in the brain's reward circuitry, correlates with memory-related theta oscillations[58]. In parallel, neuroimaging in humans suggests that RPEs correlate with increased memory-related BOLD activity[9]. In contrast, perceptual memorability is thought to engage neural activity at the junction of perception and memory in the ventral visual stream[27], independent of reward[16]. By demonstrating the positive affect regulates the RPE-memory link, and not the perception-memory link, our findings suggest that RPEs and their associated dopaminergic activity could provide a plausible neural mechanism for enhancing memory in addition to driving RL processes[59]. Linking value-based decision making to value-based memory enhancement is essential to understanding the role that dopaminergic circuits might play in memory overall.

Critically, abnormal mnemonic processes are of particular importance to mood disorders that are also linked to abnormal learning and decision-making. For example, depression is known to feature impaired processing of RL-related computations such as RPEs[22], as well as disruption of explicit memory capacity[23,60]. Acute and post-traumatic stress disorders feature pathologically strong associations with traumatic events[23] that may rely on midbrain dopaminergic modulation of synaptic plasticity[61,62]. Drug-associated cues may also develop enhanced salience in memory, contributing to substance abuse at the cost of other cues and natural rewards[63]. By taking a transdiagnostic approach to identifying a latent construct associated with mood, we demonstrate how affective state might further regulate the relationship between encoded RPEs and memory, consistent with findings from perceptual matching[64] and reward anticipation[65] studies. As such, disrupted affect could disrupt the interaction between RL and memory, distinct from impairments to RL and memory separately. These findings could have far-reaching implications for not only uncovering deeper neurocomputational mechanisms of disorders like depression and anxiety, but also suggesting that treatment and intervention strategies need to consider learning and memory deficits in an integrated fashion.

### Limitations
Our study has several limitations. First, we specifically selected images with high memorability scores to ensure that participants could perform the recognition task without needing to use reward information at all. However, more variance in the memorability scores could be helpful in establishing whether RPEs play an even larger compensatory role in recognition when perceptual information is only weakly predictive of memory. Second, we found that memory performance is more accurate when a reversal is more recent. This suggests that, while our RL model fit participant's data best, participants may also be performing hidden state inference[66] that informs their subsequent memory that might be captured by a different Bayesian model formulation than the one we utilized. Also, our analysis of psychiatric self-report and memory were exploratory and data-driven; future, pre-registered studies will be able to further investigate the influence of psychiatric symptoms on RPE-mediated memory. Finally, while memorability is thought to engage processes separate from attention[15,27] it is possible that RPE modulated memory by modulating attention. Future studies utilizing eye-tracking in concert with behavioral modeling will be best able to address this possibility.

## Conclusions

Here, we have demonstrated the interplay of perceptual information and reward information enhance memory, and how affective symptoms selectively regulate the influence of RPEs on memory. In addition to reinforcing recent work investigating the memory-enhancing effects of RPEs, these findings provide the first evidence for how value computations during learning directly interact with perceptual memorability, and how mood disorders may diminish the the beneficial effect of RPEs on memory while sparing perceptual processes. These results will thus enable future computational work investigating how models of memory may jointly and dynamically incorporate intrinsic perceptual information and extrinsic associations, and future physiological work investigating how dopaminergic circuits in the brain modulate neural activity in regions typically associated with memory.

## Data availability

The behavioral data used in this study are available at https://osf.io/awu3m/. The database used to source image stimuli is available at https://wilmabainbridge.com/facememorability2.html.

## Code availability

The analysis code and Jupyter notebooks used to generate manuscript figures are available at: https://osf.io/awu3m/.

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

## Acknowledgements

We thank Blair Shevlin and Kaustubh Kulkarni for helpful comments and suggestions. X.G. is supported by NIH R01DA043695, R21DA049243, R21MH120789, R01MH122611, R01MH123069, and R01MH124115. I.S. is supported by NIH R01MH124763. S.E.Q. is supported by NIH K99MH132873. The funders had no role in study design, data collection and analysis, decision to publish or preparation of the paper."

## Author contributions

S.E.Q. and X.G. conceived the study; S.E.Q. and A.D. analyzed the data; and S.E.Q., I.S. and X.G. wrote the paper.

## Competing interests

The authors declare no competing interests.
