## [Peer Review File · Communications Psychology]

11th Dec 23

Dear Dr Qasim,

Thank you for your patience during the peer-review process. Your manuscript titled "Positive affect modulates memory by regulating the influence of reward prediction errors" has now been seen by 3 reviewers, and I include their comments at the end of this message. They find your work of interest, but raised some important points. We are interested in the possibility of publishing your study in Communications Psychology, but would like to consider your responses to these concerns and assess a revised manuscript before we make a final decision on publication.

We therefore invite you to revise and resubmit your manuscript, along with a point-by-point response to the reviewers. Please highlight all changes in the manuscript text file.

Editorially, it is particularly important that you address the concerns from Reviewers 2 and 3 regarding other possible models or interpretations of the existing computational models. You should also address the concerns surrounding the factor analysis and robustness of the association with psychiatric symptoms highlighted by Reviewer 2. All reviewers suggest further clarity surrounding the interpretation of your results.

Please also note our requirements for statistics reporting and interpretation. Claims of no difference/no effect, or specificity for an effect in one condition require positive evidence for the absence of an effect/a difference (in the other conditions in the case of specificity claims). These can be derived from Bayesian statistics or equivalence tests. For more guidance on statistics reporting and interpretation, please consult the template and checklist linked below and ensure your revision is fully compliant.

Please note that your revised manuscript must comply with our formatting and reporting requirements, which are summarized on the following checklist:

Communications Psychology formatting checklist and also in our style and formatting guide
Communications Psychology formatting guide .

Please use the following link to submit your revised manuscript, point-by-point response to the referees' comments (which should be in a separate document to any cover letter) and the completed checklist:

[link redacted]

Please do not hesitate to contact me if you have any questions or would like to discuss these revisions further. We look forward to seeing the revised manuscript and thank you for the opportunity to review your work.

Best regards,

Patricia Lockwood

Patricia Lockwood, PhD

Editorial Board Member

Communications Psychology

orcid.org/0000-0001-7195-9559

EDITORIAL POLICIES AND FORMATTING

Editorial Policy: Policy requirements (Download the link to your computer as a PDF.)

* **CODE AVAILABILITY:** All Communications Psychology manuscripts must include a section titled "Code Availability" at the end of the methods section. In the event of publication, we require that the custom analysis code supporting your conclusions is made available in a publicly accessible repository; at publication, we ask you to choose a repository that provides a DOI for the code; the link to the repository and the DOI will need to be included in the Code Availability statement. Publication as Supplementary Information will not suffice. We ask you to prepare code at this stage, to avoid delays later on in the process.

* **DATA AVAILABILITY:**

All Communications Psychology manuscripts must include a section titled "Data Availability" at the end of the Methods section or main text (if no Methods). More information on this policy, is available at <http://www.nature.com/authors/policies/data/data-availability-statements-data-citations.pdf>.

At a minimum the Data availability statement must explain how the data can be obtained and whether there are any restrictions on data sharing. Communications Psychology strongly endorses open sharing of data. If you do make your data openly available, please include in the statement:

- Unique identifiers (such as DOIs and hyperlinks for datasets in public repositories)
- Accession codes where appropriate
- If applicable, a statement regarding data available with restrictions

- If a dataset has a Digital Object Identifier (DOI) as its unique identifier, we strongly encourage including this in the Reference list and citing the dataset in the Data Availability Statement.

We recommend submitting the data to discipline-specific, community-recognized repositories, where possible and a list of recommended repositories is provided at <http://www.nature.com/sdata/policies/repositories>.

If a community resource is unavailable, data can be submitted to generalist repositories such as figshare or Dryad Digital Repository. Please provide a unique identifier for the data (for example a DOI or a permanent URL) in the data availability statement, if possible. If the repository does not provide identifiers, we encourage authors to supply the search terms that will return the data. For data that have been obtained from publicly available sources, please provide a URL and the specific data product name in the data availability statement. Data with a DOI should be further cited in the methods reference section.

REVIEWERS' EXPERTISE:

Reviewer #1 Reinforcement learning, computational psychiatry, mood

Reviewer #2 Reinforcement learning, mental health symptoms

Reviewer #3 Reinforcement learning, depression, decision-making

REVIEWERS' COMMENTS:

Reviewer #1 (Remarks to the Author):

This study by the Qasim and colleagues seeks to determine how reward prediction errors influence memory formation. They use a simple learning task in which monetary outcomes are presented together with face stimuli which have a consistent level of “memorability”. This enabled them to determine how reward prediction errors influenced performance when given a subsequent recognition test using these face stimuli. They find that reward prediction error predicts subsequent recognition, such that stimuli paired with greater prediction errors are more likely to be recognised. They also explore how this

association is influenced by measures of positive affect, suggesting that the effect is stronger in individuals reporting higher positive affect.

The study is well conducted, using a simple but powerful design to understand how reward prediction error influences memory, and the findings represent an interesting and useful contribution to the literature. I have a few suggestions for things that could be addressed in a revision:

- 1) It looks like the outcomes were reward vs. no reward, rather than reward vs. loss – is this the case?
- 2) Using RPE as a regressor in the models makes sense intuitively, but it does leave some unanswered questions. Given that the RPE is comprised of reward and error, it's hard to be sure that it's the RPE specifically as opposed to an effect of just reward or error. It might be worth repeating the analysis including reward, absolute error, and their interaction (which essentially equates to the RPE) – this would give more clarity regarding the effect of reward and error, alongside their combination
- 3) “We next examined whether these transdiagnostic factors regulated the individual differences in the relationship between RPEs and memory (see Methods, Fig. 3B)” – I believe this should be Fig 5B, and I can't see anything in the methods about this analysis.
- 4) I don't quite understand the analysis showing an effect of positive affect on the RPE-memory relationship. The effect in the multilevel model isn't “significant”, but this seems to be followed up with a different analysis that shows it is significant? Given that the primary analysis shows a non-significant effect, this does seem a little suspicious
- 5) An alternative explanation for the results could be that it's not the RPE specifically that affects memory, but instead some kind of hidden state prediction error. In this task, RPEs are in general going to be most prominent around the reversals, but alternatively this could be a period where subjects are inferring a new hidden state to be active, which could potentially drive memory enhancement. There are a few papers now showing that models implementing this kind of hidden state inference do a good job of explaining behaviour in these sorts of tasks (e.g., Zika et al., 2023). I wouldn't necessarily want to ask the authors to test this empirically as it's probably beyond the scope of the paper, but it might be worth mentioning in the discussion.
- 6) What does the X refer to in Eq.1?
- 7) More samples could be used for the MCMC sampling – it probably wouldn't make a huge difference, but there's no reason not to go for something like 4000+ samples to potentially make the posteriors a little more stable
- 8) Which kind of orthogonal rotation was used for the factor analysis specifically?
- 9) It would be nice to include the scree plot that was used to determine the number of factors

Minor:

1) p.11 – “Baye’s” rule should be “Bayes’” rule

Reviewer #2 (Remarks to the Author):

In this study, Qasim et al., study a community sample of participants (n=206 for main effects, n=173 for relationships with psychiatric scales) to investigate a potential relationship between prediction errors experienced following a choice and memory formation. Participants perform a decision-making task followed by a memory task. The authors find that performance on the memory task is predicted, in part, by reward prediction errors, with larger reward prediction errors leading to better memory. When considering variation across psychiatric dimensions, the authors find that this relationship is stronger in people with greater positive affect.

This study examines an interesting and timely question, namely whether memory formation is influenced by the outcomes we receive from our decisions. The manuscript is well written, and the computational modelling is performed to a good standard including model parameter recovery and falsification. However, as it stands, there are still problems with the analysis and interpretation and the conclusions may not be fully supported by the data.

Major

Task: The way the two tasks were constructed, the facial stimulus that is to be remembered is completely irrelevant for the decision. This means that the question the authors are really asking becomes whether the memory of an irrelevant stimulus is influenced by the valence of a concurrent but unrelated prediction error. This makes it slightly harder to relate the process under study to real-life decisions such as those around which the manuscript is currently pitched. An explanation of whether this was an intended design choice and how it might influence the interpretation would be helpful. Also: were the participants instructed about the presence of the facial stimuli and told they will have to do a memory task on these facial stimuli in a second task, or was this not disclosed at the start of the decision task?

Modelling: From the plots in Figure 2B, there seems to be no incentive to learn anything or perform better than random in this task. From the figure, it seems that rewards did not depend on learning rate or inverse temperature which is rather counterintuitive and hard to follow given a reversal schedule was underlying the decision task. How is it possible that people who are much noisier in their choices than others or who update slowly still received the same reward on average? It would be helpful to know the

optimal learning rate - did the authors run an optimal learner model or a grid-search to check which learning rate/temperature combination would yield maximal rewards in the task?

Analysis: The absence of a relationship between RPE and PM on the one hand, and the existence of a relationship between RPE and $p(\text{hit})$ on the other hand are key to the results of this paper. Figure 2D: If the RPE is a continuous variable and a regression is fit, why is the RPE binned into terciles here? What would the result look like for a full linear fit without any binning? Figure 2E – if the RPE was binned in terciles in D, is it binned here again or used as a continuous variable now? If it was binned, please check that doing a continuous analysis would not change the conclusions. Line 94 also seems to suggest that the analysis was restricted to hits but then $p(\text{hit})$ is evaluated, which seems incompatible. Finally, it seems that maybe here the regression is fit to the data from all participants pooled which is not appropriate statistically as it treats participants as fixed effects.

Factor analysis: There are a few things that are unusual about the factor analysis: it seems the first and second factor are almost mirror images of each other. The order of the variables/questionnaire items is also not spelled out anywhere, so it is not possible to follow the factor analysis results. How correlated are the factor weights and how correlated are the derived individual factor scores (there is a supplementary figure (Figure S3), but the correlation coefficients are not included)? For a factor analysis, the recommendation is that there should be 5-10 measures per item included but there seem to be 50+ items. Thus, with only 173 participants, the factor analysis does not seem sufficiently powered. It is also not possible to follow how the factor analysis was computed because the Methods include insufficient detail - so this part of the manuscript is difficult to evaluate.

Replicability & psychiatric analyses: It would be nice to see a preregistration of the study because it seems entirely exploratory and relying on a quite small sample size of $n=206$ (or $n=173$ for cross-sectional psychiatric analyses) compared to comparable work (e.g. by Gillan et al, Wise et al. etc, Scholl et al. and others). Especially when in some of the key analyses, relationships to positive affect are only performed on a third of the data. In general, Figure 5B on which the key psychiatric conclusion hinges is problematic as the mixed effect model does not justify breaking up the positive affect score (the interaction or RPE with $f1$ scores is not significant) – the authors should also be weary of Simpson's paradox. Given the sample size of $n=173$ is already small, dividing the sample into three datasets means these correlations are not well powered.

Minor

Some important detail is missing:

- What was the point range for the RPE, was it always -100 or +100 or could it take any range within this. Or was a negative RPE just the absence of reward? If so, in Figure 2C, the +100 which is labelled as 'reward prediction error' is misleading as it should be the received reward minus the prediction (average predicted outcome).
- Figure 2E casts some doubt on the PM measure: if it is meant to index memorability, the relationship seems rather weak?
- Design: It seems important that the memorability rating is not correlated with the RPE, otherwise it is hard to disentangle the two effects. While Figure 2D is maybe meant to make this point, it would help for the actual values to be plotted here, rather than just a line of fit.
- It seems that a larger positive prediction error could mean that participants simply paid more attention – could this alternative interpretation mean that it is not directly a RPE-related process that enhances the memory?
- What was the performance on the lure trials?
- Figure S1: the title does not seem to match the reported correlation – behavioural performance does seem correlated across tasks
- Figure S4, if an absence of an effect is interpreted, appropriate Bayesian statistics in favour of the null is required but not currently shown

Reviewer #3 (Remarks to the Author):

In this paper, Qasim et al investigated how positive affect modulates memory processes through the influence of reward prediction errors. I think the authors try to address an important question and the experimental methodology is reasonably suitable for this purpose. However, I have some questions about the exact details of methodology, presentation of results etc. I communicate them below with reference to line numbers and I hope the authors will take them as constructive

The introduction to this paper is concisely and beautifully written.

Figure 1: if I am not mistaken the recognition task had 120 trials. In these trials the old faces which came from the learning part of the experiment, were they presented in the same order as participants experienced them in the learning block? Or were they randomly sorted in the recognition task. If yes,

was that randomized for each participant or pseudo-random sequence? Temporal sequence is very important in memory paradigms

In panel D, the probability of correct prediction shows 60 trials as if this is supposed to map on to the panel B from the learning experiment. I wasn't clear about this as the recognition task had 120 trials. I think the authors need to provide more information here and if needed revise the figures accordingly. If fact, they can plot experiment time courses of hits as well as misclassifications

Figure 2 in panel D the y-Axis is pm and it looks like from the shading it has a very narrow spread. However, in panel E, in the right panel the PM seems to have a much wider spread. I recommend the authors to represent this data in a heatmap grid format where X and y-Axes are pm and rpe values observed in this population and cells contain probability of correctly recognising the faces converted to heat map. That would be much more informative

Line 88 I wasn't very clear why the authors focused on only highly memorable faces only as it would be important to establish a gradient here? This should be discussed in limitations as there could be nonlinear interactions there, limiting the generalizability of results

Line 91 I am really not sure how to fit a logistic regression model if the analysis is only restricted to correct hits in the recognition task as the logistic regression model should have binary outcomes like correct versus incorrect

Line 115 what was the justification for using Zung depression scale over more commonly administered skills such as BDI or QIDS

Line 222 the authors mention of a d' score summarising the memorability of faces used in the task. How was this computed?

Equation 1: I am not 100% clear as to what "1 given subject" does in this model? Is this the model intercept?

Secondly, considering the learning phase of the task is analysed with traditional computational modelling, I think for the recognition phase of the task there is a mismatch when the authors rely on a simple/additive logistic regression model. I think at this stage the authors should try few hypothesis and construct equations as to how these variables should influence the decision value that underlies

probability of correctly recognising old faces. The paper would be especially valuable if the authors formalise few concrete hypotheses and test them through computational modelling and subsequent model comparison similar to what they provided in supplementary figure 2. Sadly, the literature is flooded with RL tasks/results as it is easy to model but only few authors actually go further construct decision models aimed at capturing other choice behaviour domains

Supplementary Figure 2, in the panel B the true parameters are they actual parameter is observed in this cohort? If not, how these values were sampled? Are these the range in which the LR and temperature parameters

I hope the authors will provide detailed revisions in response to my queries and I would be happy to review a revised version of this paper

Thank you for inviting us to submit a revision of our manuscript “Positive affect modulates memory by regulating the influence of reward prediction errors”. We appreciate the reviewers’ comments (green) and have carefully responded, point-by-point, to all suggestions (black). Blue text denotes text added to the manuscript.

Reviewer 1:

1) It looks like the outcomes were reward vs. no reward, rather than reward vs. loss – is this the case?

Yes; we now clarify this in the main text on Line 51 and the Methods on Line 256:

Each draw could result in winning either 100 points, or 0 points

2) Using RPE as a regressor in the models makes sense intuitively, but it does leave some unanswered questions. Given that the RPE is comprised of reward and error, it’s hard to be sure that it’s the RPE specifically as opposed to an effect of just reward or error. It might be worth repeating the analysis including reward, absolute error, and their interaction (which essentially equates to the RPE) – this would give more clarity regarding the effect of reward and error, alongside their combination

We followed the reviewer’s advice and repeated our main modeling analysis (Figure 3) by replacing our initial RPE linear regression with a general linear model including reward, absolute error, and their interaction. However, this GLM model performed worse than our initial model using RPE alone (Review Figure 1). Because of this, we continue to utilize RPE as a regressor; future work more specifically tailored to untangling separable reward and error contributions will be able to carefully answer this important question.

Review Figure 1: Model comparison (leave-one-out) between original regression model (“rpe_only”) and revised regression model replacing RPE with reward, absolute error, and their interaction (“reward_and_error”). Higher values indicate better models.

3) “We next examined whether these transdiagnostic factors regulated the individual differences in the relationship between RPEs and memory (see Methods, Fig. 3B)” – I believe this should be Fig 5B, and I can’t see anything in the methods about this analysis.

We thank the Reviewer for pointing out this typo. We have corrected the figure reference, and have removed the extraneous reference to the Methods section. We have instead added further detail about this analysis to that section of the Results on Lines 150-155:

Following from the trial-level results, we performed a regression analysis of subject-level memory performance as a function of participants' reliance on RPE for memory (β_{rpe}) and continuous factor scores, and found that subject's factor 1 score exhibited a significant interaction with subject's overall reliance on RPE for memory (linear-regression coefficient=1.6, $p=0.027$). This subject-level result indicated that participants who relied more on RPEs for memory also exhibited better memory overall – if they were also individuals with greater positive affect.

4) I don't quite understand the analysis showing an effect of positive affect on the RPE-memory relationship. The effect in the multilevel model isn't "significant", but this seems to be followed up with a different analysis that shows it is significant? Given that the primary analysis shows a non-significant effect, this does seem a little suspicious

Following the prior comment, we have added detail to the Results sections to better clarify the distinction between the **trial-level** multilevel model and the **subject-level** regression that follows on Lines 147-149:

While the mixed-effects model explored how factor scores influenced trial-level memory, we next examined whether subject-level individual differences in memory were explained by individual differences in factor scores and subject-level reliance on RPE for memory (β_{rpe} ; the slope of the relationship between RPE and memory fit for each subject in the mixed-effects model).

As one would expect, the positive relationship we observe for this interaction follows the same direction as the posterior for the interaction in our multilevel model for trial-level effects. We now also add a clearer interpretation of this result on Lines 153-155:

This subject-level result indicated that participants who relied more on RPEs for memory also exhibited better memory overall—if they also exhibited greater positive affect.

5) An alternative explanation for the results could be that it's not the RPE specifically that affects memory, but instead some kind of hidden state prediction error. In this task, RPEs are in general going to be most prominent around the reversals, but alternatively this could be a period where subjects are inferring a new hidden state to be active, which could potentially drive memory enhancement. There are a few papers now showing that models implementing this kind of hidden state inference do a good job of explaining behaviour in these sorts of tasks (e.g., Zika et al., 2023). I wouldn't necessarily want to ask the authors to test this empirically as it's probably beyond the scope of the paper, but it might be worth mentioning in the discussion.

Thank you for this highly insightful comment. Indeed, we had already included "time since reversal" as a predictor in the multi-level model in Figures 3 and 5, measuring the proximity, in trials, to a reversal. This analysis shows that memory performance is more accurate when a reversal was more recent (described on Line 117). We now further clarify this point in the Discussion, and add a citation to the referenced paper to improve the interpretation of this point on Line 225:

Second, we found that memory performance is more accurate when a reversal is more recent. This suggests that, while our RL model fit participant's data best, participants may also be performing hidden state inference (Zika et al., 2023) that informs their subsequent memory that might be captured by a different Bayesian model formulation than the one we utilized.

6) What does the X refer to in Eq.1?

X refers to the fixed effects utilized in the multilevel model. As stated on Line 292, these include “the following trial-level predictors: RPE, PM, and the within-block trial number (coded such that trials following a reversal restart at 1)”.

7) More samples could be used for the MCMC sampling – it probably wouldn't make a huge difference, but there's no reason not to go for something like 4000+ samples to potentially make the posteriors a little more stable

We have followed the reviewer's suggestion and MCMC sampling now utilizes 4000 tuning samples followed by 4000 draws, for a total of 8000 samples per chain. This did not alter any results, but we have altered the Methods to reflect this change.

8) Which kind of orthogonal rotation was used for the factor analysis specifically

In response to reviewer 2, we revised our factor analysis to more appropriately account for factor correlations (e.g. “mirror images”). Previously, we had used an orthogonal ‘varimax’ rotation to compute loadings, which assumes the factors are uncorrelated – an assumption which can lead to distorted loadings and reduction in explained variance. We thus switched to an oblique ‘promax’ rotation and now also compute the factor scores using the ‘tenBerge’ method appropriate for oblique rotations. This approach is thus more appropriate for our dataset and still returns similar loadings, with less mirroring between F1 and F2. **Critically, our primary results remain unchanged using this approach.** We specify these methodological details on Line 324 and 329:

We next performed factor analysis using an oblique promax rotation, using maximum likelihood estimation... We computed the factor score using the tenBerge method.

9) It would be nice to include the scree plot that was used to determine the number of factors

We have added the scree plot as a new Supplementary Figure 6 (see below):

Supplementary Figure 6: Scree plot depicting the eigenvalues for each factor extracted in the factor analysis. Eigenvalues are plotted against factor number, helping to identify the point at which adding more factors provides diminishing returns in explaining variance.

Minor:

1) p.11 – “Baye’s” rule should be “Bayes’” rule

We thank the reviewer for pointing out this typo and have corrected it.

Reviewer 2:

Task: The way the two tasks were constructed, the facial stimulus that is to be remembered is completely irrelevant for the decision. This means that the question the authors are really asking becomes whether the memory of an irrelevant stimulus is influenced by the valence of a concurrent but unrelated prediction error. This makes it slightly harder to relate the process under study to real-life decisions such as those around which the manuscript is currently pitched. An explanation of whether this was an intended design choice and how it might influence the interpretation would be helpful. Also: were the participants instructed about the presence of the facial stimuli and told they will have to do a memory task on these facial stimuli in a second task, or was this not disclosed at the start of the decision task?

Thank you for this comment. This was an intentional design choice, as the goal of this current work is to delineate the contributions of two sources of information to memory (perceptual memorability and reward prediction errors). If perceptual features of the facial stimuli were relevant to the decision, these two variables would instead share high collinearity, making it difficult to separate their respective contributions to memory. We have now revised the text in two ways to clarify this point. First, we now clarify that the use of incidental stimulus for **decision-making** was an intended design choice meant to ensure maximum dissociation between stimulus features (memorability and RPE) and decision-making in line with Wimmer et al., (2014) [PMID: 25378157] and Jang et al., (2019) [PMID: 31061490] on Lines 261-263:

Face stimuli were chosen in part to be incidental to the decision-making task in order to maximize dissociation between stimulus features and decision-making behavior (Wimmer et al., 2014, Jang et al., 2019);

Furthermore, the face stimuli were not incidental to **memory** as participants were instructed to “pay attention to the faces you see - you may need to remember them later.” We now clarify this point on Line 264:

...in contrast, participants were informed that they may need to remember the faces for a subsequent task

Second, we now improve the overall interpretability of our results by providing a clearer analogy relating our task to real-life decision making on Lines 6-8:

For example, if you choose a restaurant for dinner and unexpectedly find \$100 on the ground while eating there, does that dining experience stand out more in your memory than other restaurants?

Modelling: From the plots in Figure 2B, there seems to be no incentive to learn anything or perform better than random in this task. From the figure, it seems that rewards did not depend on learning rate or inverse temperature which is rather counterintuitive and hard to follow given a reversal schedule was underlying the decision task. How is it possible that people who are much noisier in their choices than others or who update slowly still received the same reward on average? It would be helpful to know the optimal learning rate - did the authors run an optimal learner model or a grid-search to check which learning rate/temperature combination would yield maximal rewards in the task?

1. As the reviewer requested, we performed a grid-search across learning rate and inverse temperature combinations; a combination of high learning rates and high inverse temperature yields maximal reward (as predicted by Figure 1D from Zhang et al., (2020) [PMID 32608484]). We also split the joint distribution in Figure 2A according to low vs. high reward. Together, these two plots replaced the prior plots in Figure 2B (see below), and demonstrate a) the optimal combination of parameters to maximize reward, and b) how the joint distribution of parameters from the empirical data do not approach that maxima, and thus the two distributions for low vs. high reward appear similar. To that end, we have added the following interpretation of these new plots to Lines 86-90:

We performed a grid-search over the joint parameter space, simulating actions and outcomes. The combination of parameters that maximized reward in these simulations was a combination of high learning rate and high inverse temperature (Zhang et al., 2020). Participants rarely exhibited this combination, suggesting that while they performed above chance, they rarely performed optimally; accordingly, participants' learning rate, inverse temperature, and their interaction did not show a correlation with reward accrued in the decision-making task (all $\beta < 0.03$, all $p > 0.6$; Fig. 2B).

Figure 2B: Left: simulated reward outcomes for different combinations of parameters. Warm colors denote better performance (more reward). Right: joint distribution of parameter estimates from the RW model, split by reward earned during the decision-making task (median split). Iso-proportion threshold for contours set to 0.15 for ease of visualization.

Analysis: The absence of a relationship between RPE and PM on the one hand, and the existence of a relationship between RPE and $p(\text{hit})$ on the other hand are key to the results of this paper. Figure 2D: If the RPE is a continuous variable and a regression is fit, why is the RPE binned into terciles here? What would the result look like for a full linear fit without any binning?

We apologize for the mistaken caption referring to binning – the current plot already represents the full linear fit without binning. We have included the same plot with all data points represented as reviewer Figure 2, below:

Reviewer Figure 2: Linear fit of relationship between RPE and PM ratings for each stimulus. Dots denote individual RPE and PM ratings for individual images

Figure 2E – if the RPE was binned in terciles in D, is it binned here again or used as a continuous variable now? If it was binned, please check that doing a continuous analysis would not change the conclusions.

Please see above; all statistical treatment of RPE utilized the continuous measure.

Line 94 also seems to suggest that the analysis was restricted to hits but then $p(\text{hit})$ is evaluated, which seems incompatible.

We thank the reviewer for pointing out this typo and have corrected it on Line 99-101. The analysis was restricted to ‘old’ images and not ‘novel’ images.

Finally, it seems that maybe here the regression is fit to the data from all participants pooled which is not appropriate statistically as it treats participants as fixed effects.

We now clarify on Lines 99-101 that this regression across participants is specifically meant to facilitate direct comparison with/replicate similar figures from prior work. **However, we agree with the reviewer** on this point, which is why all subsequent modeling (Figure 3, Figure 5) utilizes mixed-effects models in which participants are treated as random effects.

Factor analysis: There are a few things that are unusual about the factor analysis: it seems the first and second factor are almost mirror images of each other. The order of the variables/questionnaire items is also not spelled out anywhere, so it is not possible to follow the factor analysis results. How correlated are the factor weights and how correlated are the derived individual factor scores (there is a supplementary figure (Figure S3), but the correlation coefficients are not included)? For a factor analysis, the recommendation is that there should be 5-10 measures per item included but there seem to be 50+ items. Thus, with only 173 participants, the factor analysis does not seem sufficiently powered. It is also not possible to follow how the factor analysis was computed because the Methods include insufficient detail - so this part of the manuscript is difficult to evaluate.

To address the reviewer’s overarching concern about detail, we now include substantially more details about the factor analysis methods (Lines 314-329). We have also revised these Methods substantially to address the following point-by-point concerns:

1. We revised our factor analysis to more appropriately account for factor correlations (e.g. “mirror images”). Previously, we had used an orthogonal ‘varimax’ rotation to compute loadings, which assumes the factors are uncorrelated – an assumption which can lead to distorted loadings and reduction in explained variance. We thus switched to an oblique ‘promax’ rotation and now also compute the factor scores using the ‘tenBerge’ method appropriate for oblique rotations. This approach is thus more appropriate for our dataset and still returns similar loadings, with less mirroring between F1 and F2 (see revised Figure 4B below). **Critically, our primary results remain unchanged using this approach.** We now also provide the correlations between the factor weights on Lines 138-139:

Factor loadings for the factors were allowed to correlate ($\rho_{F1:F2}=-0.62$, $p<0.001$, $\rho_{F1:F3}=-0.22$, $p=0.1$, $\rho_{F2:F3}=-0.38$, $p=0.003$).

We also specify the correlation coefficient for the factor scores in Figure S4 (previously Figure S3), and remove the phrase ‘non-overlapping’ from our description of loadings. The bars in Figure 4B are ordered with respect to survey and ascending Factor 1 loading (now included in the figure caption), and each questionnaire item and its respective loadings is included in Table 2 in the supplement. Finally, we have added the scree plot as a new Supplementary Figure 3 to support the usage of three factors.

Figure 4B (right): Results of factor analysis using oblique 'promax' rotation indicating factor loadings across survey questions for STAI-T (blue), SDS (black) and OCI-R (red) scales. Factor 1 (left) consists of questions primarily from the anxiety and depression scales related to positive affect. Factor 2 (middle) consists of questions across all three scales related to intrusive thoughts and rumination. Factor 3 (right) consists of questions primarily from the OCD scale related to obsessive behaviors.

2. To address the reviewer’s concern that a larger sample size is needed to estimate robust factor scores, we now include an additional ~147 online participants who had completed the same exact surveys in a separate study conducted by our lab, bringing our total to n=320 for the factor analysis. **Critically, our primary results remain unchanged by including these additional participants for the factor analysis.** We describe these revised Methods on Lines 316-319:

We utilized a factor analysis to identify latent transdiagnostic structure across three surveys: the state-trait anxiety index (STAI-T), the Zung depression scale (SDS), and the obsessive-

compulsive inventory (OCI-R). These were selected to match factor analyses in prior literature (Gillan et al., 2016). We first needed to increase the sample size to ensure robust estimation of factor loadings and scores. To do so, we utilized survey data from an additional 143 online participants who had completed the same set of surveys as our task participants, bringing the total number of participants utilized for the factor analysis, specifically, to n=320.

Replicability & psychiatric analyses: It would be nice to see a preregistration of the study because it seems entirely exploratory and relying on a quite small sample size of n=206 (or n=173 for cross-sectional psychiatric analyses) compared to comparable work (e.g. by Gillan et al, Wise et al. etc, Scholl et al. and others).

We address this concern by now clarifying in the Discussion on Lines 227-228 that the psychiatric analyses were exploratory and data driven and that preregistration will lend greater rigor to our follow-up studies on the knowledge generated by our study:

Also, our analysis of psychiatric self-report and memory were exploratory and data-driven; future, pre-registered studies will be able to further investigate the influence of psychiatric symptoms on RPE-mediated memory.

However, a direct comparison in sample size between this and previous research may not be warranted as the number of survey items is substantially smaller than the referred studied (59 items across 3 surveys in this study as opposed to 209 items across 9 surveys in Gillan et al.). More importantly, we followed the reviewer's suggestion as well as previous work on adequate sampling in such analysis (i.e. 5-10 subjects per item per Costello and Osborne (2019) [DOI: <https://doi.org/10.7275/yj1-4868>]) and increased the total number of subjects for the factor analysis (new n=320). The old and new sample alone – or both samples combined – give similar factor structures. As such, we are confident in the identified factors based on results from our replication sample.

Especially when in some of the key analyses, relationships to positive affect are only performed on a third of the data. In general, Figure 5B on which the key psychiatric conclusion hinges is problematic as the mixed effect model does not justify breaking up the positive affect score (the interaction or RPE with f1 scores is not significant) – the authors should also be weary of Simpson's paradox. Given the sample size of n=173 is already small, dividing the sample into three datasets means these correlations are not well powered.

We apologize for the confusion; the key analysis relating positive affect to RPE-mediated memory **is performed on the entire dataset (n=173), not a third of the data**. Data were binned into terciles in Figure 5B purely for visualization of an interaction effect. We now clarify this important point in both the figure caption and Line 150-153:

Following from the trial-level results, we thus performed a regression analysis of subject-level memory performance as a function of participants' reliance on RPE for memory (β_{rpe}) and continuous factor scores, and found that subject's factor 1 score exhibited a significant interaction with subject's overall reliance on RPE for memory (linear-regression coefficient=1.6, $p=0.027$). This subject-level result indicated that participants who relied more on RPEs for memory also exhibited better memory overall – if they were also individuals with greater positive affect.

Minor

Some important detail is missing:

- What was the point range for the RPE, was it always -100 or +100 or could it take any range within this. Or was a negative RPE just the absence of reward? If so, in Figure 2C, the +100 which is labelled as 'reward prediction error' is misleading as it should be the received reward minus the prediction (average predicted outcome).

The RPE was computed as the difference between the outcome (+100, +0) and the expected reward, Q , estimated by the RW model. Thus, the RPE could take a continuous value within the range of -100 to $+100$. We now specify this on Line 276. We thank the reviewer for pointing out the misleading label in Figure 2C and have corrected it to be more precise.

- Figure 2E casts some doubt on the PM measure: if it is meant to index memorability, the relationship seems rather weak?

There are two main reasons why this relationship seems moderate. First, the PM measure is based on perceptual salience of the images, and does not capture other contributors to memory performance (Bainbridge et al., (2013) [PMID: 24246059]. Our study was designed to address this exact question – that how value signals such as RPE accompanied by the stimuli during encoding may also contribute to memory. Second, we constrained our stimulus set to high memorability images to ensure that participants **could** perform the memory task well without using reward information at all; as such, the range of pm scores do not represent the much wider range of pm scores in the original image set, which would demonstrate a much stronger relationship between pm scores and memory. We now clarify this point on Lines 221-223 but also cast this constrained range of memorability scores as a potential limitation:

Our study has several limitations. First, we specifically selected images with high memorability scores to ensure that participants could perform the recognition task without needing to use reward information at all. However, more variance in the memorability scores could be helpful in establishing whether RPEs play an even larger compensatory role in recognition when perceptual information is only weakly predictive of memory.

- Design: It seems important that the memorability rating is not correlated with the RPE, otherwise it is hard to disentangle the two effects. While Figure 2D is maybe meant to make this point, it would help for the actual values to be plotted here, rather than just a line of fit.

We now clearly note the Pearson' correlation used to reach this conclusion on Lines 98. We also provide Figure 2D with the actual values plotted below, for the reviewer, but do not believe this plot is easier to read/interpret than the original plot with the line of fit:

Reviewer Figure 2: Linear fit of relationship between RPE and PM ratings for each stimulus. Dots denote individual RPE and PM ratings for individual images

- It seems that a larger positive prediction error could mean that participants simply paid more attention – could this alternative interpretation mean that it is not directly a RPE-related process that enhances the memory?

Thank you for this comment. It is possible that attention subserves the observed relationship between RPEs and memory; future work utilizing eye-tracking will be able to more clearly dissociate direct RPE effects from indirect effects via attention. We now state this clearly in the Discussion, on Lines 230-232:

Finally, while memorability is thought to engage processes separate from attention (Bainbridge et al., 2017, Bainbridge et al., 2020) it is possible that RPE modulated memory by modulating attention. Future studies utilizing eye-tracking in concert with behavioral modeling will be best able to address this possibility.

- What was the performance on the lure trials?

We now plot performance across all memory trials, and overall for old vs. novel lure images in the new Supplementary Figure 2 as seen below and we now discuss this result on Lines 70-71:

Supplementary Figure 2: A) Overall recognition memory performance, indexed by the probability of either a successful hit or a correct rejection of a novel lure as a function of trial. Shaded lines denote 95% confidence interval.} B) Memory performance split by image type (old images vs. novel lure images). For old images, $p(\text{recognition})$ is equivalent to $p(\text{hit})$. For novel lure images, $p(\text{recognition})$ is equivalent to $p(\text{correct reject})$. Error bars denote standard error.

- Figure S1: the title does not seem to match the reported correlation – behavioural performance does seem correlated across tasks

We thank the reviewer for pointing out this error and have corrected it.

- Figure S4, if an absence of an effect is interpreted, appropriate Bayesian statistics in favour of the null is required but not currently shown

We now provide statistics in favor of the null hypotheses depicted in Figure S4 (now Figure S5 in the revised manuscript, on Line 160).

Reviewer 3:

Figure 1: if I am not mistaken the recognition task had 120 trials. In these trials the old faces which came from the learning part of the experiment, were they presented in the same order as participants experienced them in the learning block? Or were they randomly sorted in the recognition task. If yes, was that randomized for each participant or pseudo-random sequence? Temporal sequence is very important in memory paradigms

Faces were randomly sorted in the recognition experiment for each participant. We now clarify this on Line 266.

In panel D, the probability of correct prediction shows 60 trials as if this is supposed to map on to the panel B from the learning experiment. I wasn't clear about this as the recognition task had 120 trials. I think the authors need to provide more information here

and if needed revise the figures accordingly. In fact, they can plot experiment time courses of hits as well as misclassifications

We now clarify in Figure 1's caption that this plot shows the probability of **subsequent recognition** for each of the 60 trials from the learning experiment (as the reviewer proposed, this is indeed meant to map to panel B). The x-axis for both panels has accordingly also been re-labelled to "2-arm bandit trials" to make this clearer. However, we have also followed the reviewer's advice to plot all memory outcomes as a function of **memory trial** and have also separated out overall performance for old vs. novel lure images and included this as a new Supplementary Figure 2, seen below, and discuss this result on Lines 70-71:

...though memory performance tended to decrease towards the end of the recognition period and was asymmetric between old and novel lure images ($z=31.8$, $p<0.001$; Fig. S2A-B).

Supplementary Figure 2: A) Overall recognition memory performance, indexed by the probability of either a successful hit or a correct rejection of a novel lure as a function of trial. Shaded lines denote 95% confidence interval.} B) Memory performance split by image type (old images vs. novel lure images). For old images, $p(\text{recognition})$ is equivalent to $p(\text{hit})$. For novel lure images, $p(\text{recognition})$ is equivalent to $p(\text{correct reject})$. Error bars denote standard error.

Figure 2 in panel D the y-Axis is pm and it looks like from the shading it has a very narrow spread. However, in panel E, in the right panel the PM seems to have a much wider spread. I recommend the authors to represent this data in a heatmap grid format where X and y-Axes are pm and rpe values observed in this population and cells contain probability of correctly recognising the faces converted to heat map. That would be much more informative

We have followed the reviewer's advice to plot a heatmap of hit probability as a function of rpe and pm values (see new Supplementary Figure 3, below). We now reference this joint memory space on Lines 106-107:

The joint relationship between image features and hit probability recapitulated these findings (Fig. S3).

Supplementary Figure 3: Heatmap depicting the probability of a correct recognition (hit) across all trials and participants, as a function of both image RPE and image PM. Warm colors denote better performance (more hits).

Line 88 I wasn't very clear why the authors focused on only highly memorable faces only as it would be important to establish a gradient here? This should be discussed in limitations as there could be nonlinear interactions there, limiting the generalizability of results

We selected high memorability images to ensure that participants **could**, in theory, perform the memory task well without using reward information at all (Line 222). However, we acknowledge the reviewer's point that this could be a potential limitation, and now clearly state so on Lines 222-223:

Our study has several limitations. First, we specifically selected images with high memorability scores to ensure that participants could perform the recognition task without needing to use reward information at all. However, more variance in the memorability scores could be helpful in establishing whether RPEs play an even larger compensatory role in recognition when perceptual information is only weakly predictive of memory.

Line 91 I am really not sure how to fit a logistic regression model if the analysis is only restricted to correct hits in the recognition task as the logistic regression model should have binary outcomes like correct versus incorrect

We thank the reviewer for pointing out this typo. This sentence was corrected on Lines 99-101:

First, we assessed the probability of hits, or successful recognition of old images during the recognition memory task across all participants...

Line 115 what was the justification for using Zung depression scale over more commonly administered skills such as BDI or QIDS

We now clarify that the choice of depression scale was in order to align with prior factor analysis work done by Gillan et al (2016), on Lines 315-317:

We utilized a factor analysis to identify latent transdiagnostic structure across three surveys: the state-trait anxiety index (STAI-T), the Zung depression scale (SDS), and the obsessive-

compulsive inventory (OCI-R). These were selected to match factor analyses in prior literature (Gillan et al., 2016).

Line 222 the authors mention of a d' score summarising the memorability of faces used in the task. How was this computed?

We now specify the computation for d' on Lines 268-269 in the Methods section:

We computed d' , a signal-detection metric, for each subject by subtracting the z-score corresponding to the false-alarm rate from the z-score corresponding to the hit rate (Wick and Norman, 1966).

Equation 1: I am not 100% clear as to what "1 given subject" does in this model? Is this the model intercept?

We now clarify that this term refers to the random intercept (e.g. one intercept for every subject) on Lines 295-297:

The random effects allow the influence of RPE and PM to vary across subject, as well allowing for a random intercept such that one intercept is fit per subject.

Secondly, considering the learning phase of the task is analysed with traditional computational modelling, I think for the recognition phase of the task there is a mismatch when the authors rely on a simple/additive logistic regression model. I think at this stage the authors should try few hypothesis and construct equations as to how these variables should influence the decision value that underlies probability of correctly recognising old faces. The paper would be especially valuable if the authors formalise few concrete hypotheses and test them through computational modelling and subsequent model comparison similar to what they provided in supplementary figure 2. Sadly, the literature is flooded with RL tasks/results as it is easy to model but only few authors actually go further construct decision models aimed at capturing other choice behaviour domains

We followed the reviewer's advice and expanded our computational modeling to the recognition phase. We utilized **drift-diffusion** models to test the contribution of RPE and PM to drift rate during memory search. The results of this analysis provide further support for the conclusions from Figure 3, and we have added the following figure as a new Supplementary Figure 5 (see below). We also now detail this new modeling approach to the recognition phase of the task in the Results on Lines 119-126 and the Methods on Lines 278-287:

Results:

To examine how RPE and PM influence a different index of memory behavior, we utilized drift-diffusion models (DDM) fit to participants reaction time and choices during the recognition memory task (Fig. S5A). Specifically, we assessed whether RPE or PM more strongly modulated drift rate (Fig. S5 B) - if either RPE or PM upregulated drift rate, it would suggest that this feature contributes positively to evidence accumulation in support of the recognition of the target image in opposition to evidence accumulating against it. Both RPE (posterior mean=0.046) and PM (posterior mean=0.036) contributed positively to drift rate, but the model integrating RPE was preferred to the model integrating PM (Fig. S5 C), suggesting that RPE explained more variance in participant recognition responses and reaction time than PM.

Methods:

To model the influences of RPE and PM on memory search during recognition, we utilized drift-diffusion models (DDMs), which fit a noisy sequential sampling process to choice data such that relative evidence is accumulated over time until reaching a decision boundary (e.g. a recognition choice) (Ratcliff, 1978). We first excluded reaction times beyond 3 standard deviations away from the subject-level mean, and/or those shorter than 300 ms or exceeding 10 seconds. Then, in two separate hierarchical DDMs, we modeled drift rate (v), the rate of evidence accumulation prior to making a recognition choice, as a function of RPE or PM, with subject as a random effect:}

$$v \sim \text{RPE} + (1|\text{subject}) + (\text{RPE}|\text{subject})$$

$$v \sim \text{PM} + (1|\text{subject}) + (\text{PM}|\text{subject})$$

The remaining free model parameters, including non-decision time (t), starting point (z), and boundary separation (a) were fit with complete pooling across participants. We utilized a hierarchical approach due to relatively low number of trials contributed by each participant (Wiecki et al., 2013).

Supplementary Figure 5: A) Log reaction times for old images during the recognition memory task, split by hits (dark) and misses (light).

B) Graphical model of DDM framework. The influence of image features on drift rate (v) were separately tested in individual models that partially pooled the influence of either RPE (purple) or PM (green).

C) Model comparison of the two DDMs showing that the model assessing the influence of RPE on drift rate fit the behavioral data better than the model assessing the influence of PM on drift rate.

D) Parameter recovery demonstrates good recovery of primary DDM parameters for the RPE-based DDM model.

Supplementary Figure 2, in the panel B the true parameters are they actual parameter is observed in this cohort? If not, how these values were sampled? Are these the range in which the LR and temperature parameters

We now clarify that we utilized the actual parameters observed in this cohort for the true parameters on Line 310:

...we used each model to simulate behavior for 206 agents utilizing the true parameters sampled from our cohort.

26th Feb 24

Dear Dr Qasim,

Thank you for your patience during the peer-review process. Your manuscript titled "Positive affect modulates memory by regulating the influence of reward prediction errors" has now been seen by the same 3 reviewers, and I include their comments at the end of this message.

The reviewers find your work improved, but also point to some significant shortcomings in the revised version. We remain interested in the possibility of publishing your study in *Communications Psychology*, but would like to consider your responses to these concerns and assess a revised manuscript before we make a final decision on publication.

We therefore invite you to revise and resubmit your manuscript, along with a point-by-point response to the reviewers. Please highlight all changes in the manuscript text file.

Editorially, we emphasize that the revision must engage with each of the Reviewer #2 and Reviewer #3's methodological concerns by presenting stronger evidence and - where appropriate - additional analyses to support the key claims. Only revising the limitations section, while necessary where additional revisions cannot resolve all ambiguities, is not sufficient to alleviate these concerns.

I am attaching an Editorial Requests Table that details critical reporting requirements for the revised manuscript. Please attend to each item and ensure your manuscript is fully compliant. We are requesting that your manuscript aligns with these requirements as this facilitates the evaluation of your manuscript, reducing delays in re-review and potential future acceptance. If your revised manuscript is not aligned with these requests on major issues, such as those concerning statistics, it may be returned to you for further revisions without re-review. Additional information can be found in our style and formatting guide *Communications Psychology* formatting guide.

Please use the following link to submit your

- revised manuscript,
- point-by-point response to the referees' comments,

- cover letter (as a separate document),
- the Editorial Policy Checklist (see below),
- the Reporting Summary (see below), and
- the completed Editorial Request Table (attached):

[link redacted]

Best regards,

Patricia Lockwood

Patricia Lockwood, PhD

Editorial Board Member

Communications Psychology

orcid.org/0000-0001-7195-9559

REVIEWER REPORTS:

Reviewer #1 (Remarks to the Author):

The reviewers have done a thorough job of addressing my comments and I have nothing further to suggest.

Reviewer #2 (Remarks to the Author):

The authors have done a great job addressing my concerns and I believe the manuscript is clearer and stronger as a result. The replies to the reviewer's points have, however, raised a few additional concerns.

Factor analysis: The first two factors are highly correlated (individual factor scores: -0.71), and even factors 2 and 3 have a correlation of -0.58 . Thus, including all three factor scores simultaneously in the same model can lead to flawed results. The analysis shown in Figure 5 relating the transdiagnostic constructs to behaviour should be repeated with just f1, or just f2, or just f3 included in the model to verify which results hold/replicate in this way. These three individual terms (and their interactions with RPE) will be too correlated to allow them to compete for variance (having them compete can artificially shift parameter estimates in the same direction given their negative correlation). It is also noticeable and worrying that the effect of f1 flipped from being negative to being positive in Figure 5A in terms of its effect size when comparing the original and the revised manuscript (same for f2 x rpe) - despite the authors reporting their conclusions remained unchanged.

As an aside, it is disappointing that the authors, when probed about correlations between factors, did not directly paste the relevant values (correlations between individual factor scores) into the reviewer's reply. Instead, they referred to other irrelevant values (correlation between factor weights) and the wrong Supplementary Figure (Figure S4) - the actual values are in Figure S7.

Modelling: It is unfortunate that the optimal parameters for alpha and beta were not investigated at the time of generating the schedule. The optimal learning rate is almost 1 which means people would do quite well (in terms of reward) to just follow a win-stay, lose-shift rule, which is a much less interesting and quite basic heuristic that does not rely on computing prediction errors. To write a convincing

manuscript about RPEs, it seems important to show that RPEs truly influenced behaviour. Can the authors show that a win-stay/lose-shift rule does not explain behaviour better?

I also still do not understand why the new plot in Figure 2B on the right does not reflect the reward distribution of the optimal learner on the left. The people in the top right (close to $\alpha = 1$ and with the highest beta) should be the ones getting the most reward according to the simulations on the left in 2B. Can you explain why this is not the case? Would a clearer difference come out when focusing on the reward gained in the half of trials when learning has stabilized following a reversal compared to immediately after reversal (i.e., trials 7-12 and 18-24 and 30-36 and 42-48 in the schedule in Figure 1B)?

Analysis: Does the result treating participants as fixed-effects hold when treating them as mixed-effects (line 99-101)? Even if previous studies used an (inappropriate) fixed-effects design, this does not justify repeating the same (inappropriate) analysis here.

Task: Thanks for clarifying that the incidental nature of the faces was an intentional design choice. The instruction to the participant, "You may need to remember them later" is quite vague. Were participants incentivised in any way to remember the faces, i.e., were they paid more for better performance on the memory task?

Reviewer #3 (Remarks to the Author):

[comments on the Revised version]

I would like to thank the authors for responding to my queries. I also have to say that some of the things that I picked up in the original submission remained the same in the manuscript and only supplementary figures are added.

For example, Figure 2D-E where there is a mismatch between pm range on different axis remains the same, which I found bit puzzling. I liked the newly generated heatmap (SFig3) the authors presented but surely this must be generated by a function, I do not think that the experiments would have such a fine gradient (I sincerely apologise if I underestimated the gradient in their raw data and I am wrong). I would appreciate it if the authors rely on the raw data, whatever the range/coverage of the raw data would be (there might be NaN cells, combinations which are not covered in the experiment) and report this as a main figure.

I was also intrigued by author's choice of drift diffusion for modelling the recognition phase. Initially, I was skeptical, highlighting that there was a mismatch between traditional computational modelling for the learning phase and an additive logistic regression model for the recognition phase, and I wanted the authors to test few hypothesis about the exact relationship between rpe and PM and their influence on choice through traditional computational modelling. Now, the current drift modelling, although interesting, is a similar additive test in a multiple linear regression where the data now is the drift rate. Furthermore, drift models are also utilized in perceptual/evidence accumulation tasks like streaming for information. The current task has a strong memory component that should be triggered by visual features of the faces, and the results suggest that RPE which is a secondary process to PM in terms of evidence accumulation has greater impact on the drift rate (to me, a rather surprising finding).

Here, what the reader should think is happening? participants streaming information from memory?

I think modelling should provide clarity on relationships which cannot be revealed by simple statistical analyses.

Also, rpe and PM can easily influence other parameters of the drift model, easiest prediction would be the non-decision time which might truncate for higher RPE or PM. I really don't want to create additional obstacles/burden to the authors, but I hope they can also see that uncorrected relationship testing here remains bit problematic.

I really want to leave it to the authors here, as it is a late stage in the revision process. They might want to construct/test few non-additive hypotheses to unpack how rpe and pm (and perhaps their interaction) influence recognition decisions, or they can also write a detailed limitations sections overlapping with some of the issues that I highlighted above.

I hope the authors find my comments useful.

EDITORIAL POLICIES

We ask that you ensure your manuscript complies with our editorial policies and reporting requirements.

To that end, we require revised manuscripts to be accompanied by two completed items: a reporting summary that collects information on study design and procedure, and an editorial policy checklist that verifies compliance with all required editorial policies.

Nature Research Reporting Summary

Editorial Policy Checklist

All points on the policy checklist must be addressed. Your revised manuscript can only be sent back to the referees if these checklists are completed and uploaded with the revision.

Notes: If you have submitted a Stage 1 Registered Report, Review, Primer, Comment, or Perspective you do not need to submit these forms. If you have already submitted these forms, you may disregard this request.

We thank the editor and reviewers for their constructive comments on the revised manuscript. We have made substantial revisions and performed additional analyses to address the Reviewer's suggestions. These are detailed in the point-by-point responses below (reviewer comments are in green, responses in black, and edited text in blue). All revisions in the revised manuscript are also highlighted in blue for ease in identifying changes.

Reviewer #2:

The authors have done a great job addressing my concerns and I believe the manuscript is clearer and stronger as a result. The replies to the reviewer's points have, however, raised a few additional concerns.

We thank the Reviewer for their prior comments, which helped strengthen the manuscript.

Factor analysis: The first two factors are highly correlated (individual factor scores: -0.71), and even factors 2 and 3 have a correlation of -0.58. Thus, including all three factor scores simultaneously in the same model can lead to flawed results. The analysis shown in Figure 5 relating the transdiagnostic constructs to behaviour should be repeated with just f1, or just f2, or just f3 included in the model to verify which results hold/replicate in this way. These three individual terms (and their interactions with RPE) will be too correlated to allow them to compete for variance (having them compete can artificially shift parameter estimates in the same direction given their negative correlation).

We have followed the Reviewer's suggestion to conduct three separate mixed-effects linear models utilizing **either f1, or f2, or f3** alone. Model comparison revealed that the model using f1, the factor score measuring positive affect, performed best, as seen in the new Figure 5A (below). The regression weights produced by the f1-only model (β_{rpe} , β_{pm}) strengthened our finding that "participants who relied more on RPEs for memory also exhibited better memory overall—if they were also individuals with greater positive affect" from $p=0.027$ to $p=0.011$ (new Figure 5B, below). We are grateful that the Reviewer noticed the issue of multicollinearity, as their suggestion greatly improved this analysis and relevant results.

Figure 5: **Mood regulates memory-enhancing effect of RPEs.** A) Top: Model comparison of the three mixed-effects regression models showing that the model assessing the influence of factor 1 on memory fits the empirical data better than the model assessing the influence of factors 2 or 3. Gray triangle

indicates difference in WAIC scores. Bottom: Posterior distributions for fixed effects in mixed-effects model examining how transdiagnostic factor score f1 impacts memory. Shaded portion represents the 95% high-density (HDI) interval. Vertical line indicates a coefficient of 0. Posterior distributions that include 0 are shaded gray, while those that do not are shaded red, indicating a meaningful effect. B) Top: relationship between β_{rpe} , (subject-level random effect from the mixed-effects model) and memory performance, organized by a tercile split (for visualization only) of positive affect (measured by factor 1 score). Asterisk indicates significant interaction between factor score and β_{rpe} , in predicting memory performance (linear-regression coefficient=1.4, $p=0.011$). Bottom: relationship between β_{pm} , (subject-level random effect from the mixed-effects model) and memory performance, arranged by a tercile split of positive affect (measured by factor 1 score). Dots denote values for individual participants. Solid line indicates linear model fit to participant data.

We have now also revised the manuscript on Lines 139-155 as below:

We next sought to understand how subject level factor scores (f1, f2, f3) predicted trial-level memory performance. Because these three factors were correlated, we utilized three separate Bayesian mixed-effects models and performed model comparison to identify which of these factors explained the most variance in memory performance (see *Methods*). The model including factor 1 (positive affect) performed marginally better than the models including factor 2 (intrusive thoughts and rumination) or factor 3 (obsessive-compulsive behaviors; Fig. 5A). Subjects' factor 1 score did not modulate trial-level memory performance, or interact with trial-level RPE (main effect 95% HDI include 0; Fig. 5A). Because the model including only factor 1 performed best, we used the regression coefficients from this mixed-effects model for subsequent analyses.

While the mixed-effects model explored how factor scores influenced trial-level memory, we next examined whether subject-level individual differences in memory were explained by individual differences in factor scores and subject-level reliance on RPE for memory (β_{rpe} ; the slope of the relationship between RPE and memory estimated for each subject in the mixed-effects model). Following from the trial-level results, we thus performed a regression analysis of subject-level memory performance as a function of participants' reliance on RPE for memory (β_{rpe}) and continuous factor scores, and found that subject's factor 1 score exhibited a significant interaction with subject's overall reliance on RPE for memory (linear-regression coefficient=1.4, $p=0.011$). This subject-level result indicated that participants who relied more on RPEs for memory also exhibited better memory overall – if they were also individuals with greater positive affect.

Using the new regression weights from the f1-only model, f2- and f3-scores continued to demonstrate no influence on the relationship between β_{rpe} and memory, while participants who relied more on PM for memory now exhibit better memory overall if they were individuals with higher anxiety and rumination (interaction of f2 score and β_{pm} , $p=0.015$). We now note this new finding on Lines 159-164 as follows and have revised Supplemental Figure 8 to reflect this new result, but it does not alter our primary results or interpretations.

In addition, we found that subjects' factor 2 score exhibited a significant interaction with subjects' overall reliance on PM for memory (linear-regression coefficient=12.8, $p=0.015$) suggesting that participants who relied more on perceptual information for memory exhibited better memory

overall – if they were more anxious and ruminative (higher f2 scores).

Figure S8: Intrusive thoughts and rumination (f2) and obsessive-compulsive behaviors (f3) do not regulate memory-enhancing effects of RPEs.

It is also noticeable and worrying that the effect of f1 flipped from being negative to being positive in Figure 5A in terms of its effect size when comparing the original and the revised manuscript (same for f2 x rpe) - despite the authors reporting their conclusions remained unchanged.

The changes the Reviewer describes here could emerge from the inclusion of correlated factors scores as multicollinear predictors in the mixed-effects model. Removing the correlated predictors (as in new Figure 5A, above), resulted in f1 weight being negative, as in the original manuscript (which used uncorrelated factor scores to begin with). Our primary conclusions remained largely unchanged because they only utilized the regression weights for RPE (β_{rpe}) from the model (Figure 5B), not the regression weights for any of the factor scores.

As an aside, it is disappointing that the authors, when probed about correlations between factors, did not directly paste the relevant values (correlations between individual factor scores) into the reviewer's reply. Instead, they referred to other irrelevant values (correlation between factor weights) and the wrong Supplementary Figure (Figure S4) - the actual values are in Figure S7.

We apologize that the text of our prior reply mistakenly referred to the incorrect values and Supplementary Figure. These are the correct correlation values as shown in Fig S7: (f1 vs f2: $\rho=0.71$, f1 vs f3: $\rho=-0.36$, f2 vs f3: $\rho=0.57$).

Modelling: It is unfortunate that the optimal parameters for alpha and beta were not investigated at the time of generating the schedule. The optimal learning rate is almost 1 which means people would do quite well (in terms of reward) to just follow a win-stay, lose-shift rule, which is a much less interesting and quite basic heuristic that does not rely on computing prediction errors. To write a convincing manuscript about RPEs, it seems important to show that RPEs truly influenced

behaviour. Can the authors show that a win-stay/lose-shift rule does not explain behaviour better?

We agree with the Reviewer – in the future, it is crucial to simulate optimal parameter spaces prior to empirical data collection. We also want to clarify that the manuscript already contains the requested comparison between the RL model and a win-stay/lose-shift model (Lines 78-82):

In addition, we tested this model against alternative models which do not rely on cached value or RPEs, including a heuristic win-stay, lose-shift model and a Bayesian filter model estimating the probability of reward for correct choices as well as the probability of reward reversal (Table 1). We performed model comparison (see *Methods*) to select the model that provided the best and most parsimonious fit for the majority of participants' data. The winning model was the Rescorla-Wagner (RW) model ($\chi^2=18.5$, $p<0.001$, χ^2 test of proportions, Fig. S4 A-C)

We apologize for the lack of clarity; we now include further details on these alternative models in the *Methods* on Lines 288-297:

In addition to the Rescorla-Wagner model, which caches values for trial-by-error decision-making, we also constructed alternative models to capture heuristic switching behavior and Bayesian estimation of task reward state. In the heuristic model, agents keep selecting a choice until they lose, at which point they shift to the other choice, with one free parameter (ϵ) capturing choice bias (Table 1). The Bayesian filter model is based on two hidden states: one in which the purple deck is the correct choice, and the other in which the orange deck is the correct choice, with some probability that states have reversed on each trial. The model computes the likelihood that a choice is correct or incorrect as a function of inferred probability of reward for the current state. Action probabilities are computed from this likelihood, taking into account the inferred probability that a state switch (e.g. a reward reversal) has occurred (Eckstein et al., 2022). The free parameters for this model are the probability of reward, and the probability of reversal (Table 1).

I also still do not understand why the new plot in Figure 2B on the right does not reflect the reward distribution of the optimal learner on the left. The people in the top right (close to $\alpha = 1$ and with the highest beta) should be the ones getting the most reward according to the simulations on the left in 2B. Can you explain why this is not the case? Would a clearer difference come out when focusing on the reward gained in the half of trials when learning has stabilized following a reversal compared to immediately after reversal (i.e., trials 7-12 and 18-24 and 30-36 and 42-48 in the schedule in Figure 1B)?

We apologize for the confusion – because the variance in empirical parameters and reward was much smaller than the variance in our simulations, a simple median split on the empirical reward distribution was not sufficient to illustrate differences in reward between participants with different RL parameters. We now take the more logical, inverse approach by comparing rewards between groups that are maximally distant in parameter space. To do this we performed **quantile** splits to identify participants with the largest combination of RL parameters ($n=20$) and the smallest combination of RL parameters ($n=22$), leaving out the middle quantiles to maximize distance in parameter space. The large distance between these groups in parameter space is now indicated the new Figure 2B (below, left). We confirmed the Reviewer's hypothesis that this difference manifests as a function of learning stabilization, finding that the two sub-populations appear to diverge in performance as a function of early vs. late trials within blocks (new Figure 2B, top right), with participants closer to the top right of the parameter space accumulating more reward than those closer to the bottom left during the second half of each block (z-test of proportions $z=2.0$, $p=0.047$, new Figure 2B, bottom right). We have added this analysis to the manuscript as below on Lines 89-92 and revised Figure 2B (below) to reflect these findings, and thank the Reviewer for their helpful suggestions.

Accordingly, a comparison between the subset of participants with the highest combination of these parameters vs. the lowest combination (categorized by quantile split) illustrated a dissociation in optimal behavior during the first and second half of each block (Fig. 2B), resulting in higher reward for the participants with the higher combination of learning rate and inverse temperature after learning had stabilized within each block ($z=2.0$, $p=0.047$; Fig. 2C).

Figure 2B: Left: simulated reward outcomes for different combinations of parameters. Warm colors denote better performance (more reward). Inset boxes indicate subset of participants with larger parameters (black, $n=20$), and smaller parameters (red, $n=22$). Marginal distributions of parameters are indicated for each subset. Right, top: logistic fit of reward outcome as a function of trial within each block (prior to reversal), for participants with larger parameters (black) vs smaller parameters (red). Right bottom: Comparison of total reward accumulated for participants during the second half of each block with larger (black) vs smaller (red) parameters, Asterisk indicates significant difference ($z=2.0$, $p=0.047$). Dots indicate mean performance for individual subjects.

Analysis: Does the result treating participants as fixed-effects hold when treating them as mixed-effects (line 99-101)? Even if previous studies used an (inappropriate) fixed-effects design, this does not justify repeating the same (inappropriate) analysis here.

We apologize for the confusion – a majority of the manuscript already relies on mixed-effects modeling (and the results do hold). The mixed-effects regression framework is described throughout the manuscript (including Figures 3 and 5), and in the *Methods* in the section titled “Bayesian mixed-effects regression”. To improve clarity, we now remove the lines highlighted by the reviewer (previously Lines 99-101) and moved the old Figure 2E (with the group-level and subject-level regression fits) to the supplement (now Figure S4 purely for visualization) so we can more clearly emphasize the mixed-effects regression for readers. See new Lines 102-108 below:

The group-level and subject-level relationships between RPE and memory and PM and memory replicated prior studies investigating the effects of these individual features on hit probability (Davis et al., 2016, Rouhani et al., 2018, Jang et al., 2019, Calderon et al., 2021) (Fig. S4). We next sought to understand the parallel contributions of RPE and PM to memory beyond the probability of hits alone, while simultaneously accounting for subject-level RL parameters, demographics, and random-effects. To this end, we utilized a Bayesian mixed-effects logistic regression model to measure the importance of extrinsic RPE information and intrinsic perceptual information to correct vs. incorrect memory performance accounting for all four types of memory responses (hits, correct rejections, misses, and false alarms; see *Methods*).

Task: Thanks for clarifying that the incidental nature of the faces was an intentional design choice. The instruction to the participant, “You may need to remember them later” is quite vague. Were participants incentivised in any way to remember the faces, i.e., were they paid more for better performance on the memory task?

We now clarify that participants were not incentivized to remember faces, and were only paid for performance in the decision-making task, on Lines 274-276 as follows:

Participants were only paid for their performance in the decision-making task, however, to ensure there was no direct, instructed link between memory performance and reward attainment.

Reviewer #3 (Remarks to the Author):

I would like to thank the authors for responding to my queries. I also have to say that some of the things that I picked up in the original submission remained the same in the manuscript and only supplementary figures are added. For example, Figure 2D-E where there is a mismatch between pm range on different axis remains the same, which I found bit puzzling.

We apologize for this confusion – these axes have different ranges because the raw values represent fundamentally distinct metrics (population-normed memorability vs reward-prediction error). These metrics are z-scored in all linear models, so we now also z-score these values to provide matched axes and scales (below). Note the narrow range for the PM data in panel D; as we mention on Lines 98-99, we purposefully selected images from a narrow range of PM scores to limit our stimuli to those that are considered “highly memorable”. Additionally, in response to Reviewer 2 we have moved Figure 2E to the supplement (now Figure S4).

Figure 2D: Relationship between RPE and PM ratings for each stimulus. Dark line indicates model fit to all participants' data. Light lines indicate models fit to individual participants' data.

Figure S4: Logistic fit of probability of hits as a function of RPE magnitude (left) and PM (right). Dark line indicates model fit to all participants' data. Light lines indicate models fit to individual participants' data. Model coefficients denoted above. Coefficients indicate strength of fixed effects logistic-regression and asterisks indicate significance (both $p < 0.001$)

I liked the newly generated heatmap (SFig3) the authors presented but surely this must be generated by a function, I do not think that the experiments would have such a fine gradient (I sincerely apologise if I underestimated the gradient in their raw data and I am wrong). I would appreciate it if the authors rely on the raw data, whatever the range/coverage of the raw data would be (there might be NaN cells, combinations which are not covered in the experiment) and report this as a main figure.

We have added the unsmoothed heatmap to the Figure as requested, and have also moved the heatmaps to Figure 2E in the main manuscript. In addition, we revised the bins used to generate the heatmap to ensure matched axes and scales (z) as done above for Figure 2D.

Figure 2E: Heatmap depicting the probability of a correct recognition (hit) across all trials and participants, as a function of both image RPE and image PM. Warm colors denote better performance (more hits).

We refer to these results on Lines 99-101:

We first confirmed that PM ratings and model-estimated RPEs were orthogonal ($\rho=-0.02$, Fig. 2D) and plotted their joint contribution to the probability of correct recognition (Fig. 2E) to visualize their relative contribution of both streams of information.

I was also intrigued by author's choice of drift diffusion for modelling the recognition phase. Initially, I was skeptical, highlighting that there was a mismatch between traditional computational modelling for the learning phase and an additive logistic regression model for the recognition phase, and I wanted the authors to test few hypothesis about the exact relationship between rpe and PM and their influence on choice through traditional computational modelling. Now, the current drift modelling, although interesting, is a similar additive test in a multiple linear regression where the data now is the drift rate.

We now test several alternative DDMs to explore non-additive relationships (e.g. polynomial influence of RPE, which may feature equivalent influence for positive and negative RPEs, and logarithmic influence of RPE) between drift rate and RPE, and find that our original model best describes the data (new Figure S5E, below):

Figure S5E: Model comparison of alternative DDM models testing non-linear relationships between drift rate and RPE, showing that the model using a simple linear relationship fits the empirical data best. Gray triangle indicates difference in WAIC scores.

We now describe this result on Lines 124-126:

We computed several alternative models testing non-linear (e.g., logarithmic and polynomial) relationships between RPE, PM and drift-rate; however the linear model fit the behavioral data best.

Broadly, we believe that our use of drift-diffusion modeling to unveil specific latent cognitive mechanisms that RPE could modulate (e.g. rate of evidence accumulation) directly addresses the Reviewer's overall desire to see non-RL computational models used to glean insight about recognition behavior. However, if the Reviewer had other non-RL and non-DDM models in mind, it would be helpful if they provide specific suggestions so we may try those approaches within project scope.

Furthermore, drift models are also utilized in perceptual/evidence accumulation tasks like streaming for information. The current task has a strong memory component that should be triggered by visual features of the faces, and the results suggest that RPE which is a secondary process to PM in terms of evidence accumulation has greater impact on the drift rate (to me, a rather surprising finding). Here, what the reader should think is happening? participants streaming information from memory?

We thank the Reviewer for suggesting modeling of memory behavior in their prior review, as this allowed us to estimate the surprising influence that RPE has on drift rate compared to PM. We now lead with a helpful interpretation in the Introduction on Line 40-42 as follows:

Participants better remembered stimuli associated with more positive RPEs and those with higher perceptual memorability ratings, in line with prior findings (Bainbridge et al, 2013, Davis et al., 2016, Jang et al., 2019), but positive RPEs provided a faster source of meaningful evidence for participant's successful recognition of prior stimuli.

To help readers understand the importance of this result we have added further interpretation to the Discussion of how our finding might reflect the influence of reward circuitry in evidence accumulation in memory (as opposed to the influence visual features might have on evidence accumulation during perceptual tasks) on Lines 190-196 as follows:

While mixed-effects modeling demonstrated that both positive RPE and high PM contributed to successful memory, drift-diffusion modeling of reaction times during memory revealed that positive RPEs more meaningfully up-regulated drift rate during memory search. While the strength of visual information is implicated in evidence accumulation in perceptual decision-making (Palmer et al., 2005), our results suggest that the reward computations associated with surprising rewards provide more important evidence per unit of time for matching the recognition cue stimulus to the image stored in memory. This finding provides further support for a functional dissociation between RPE- and perceptually-mediated memory enhancement.

We have also added an explanatory DDM figure to the main manuscript in the new Figure 3B, C, below:

B)

C)

Figure 3: B) Top: Schematic of drift-diffusion models of recognition reaction time and choice that allow RPE (purple) or PM (green) to influence the drift rate (v). Bottom: Model comparison of the RPE and PM DDM models, showing that the model using RPE fits the empirical data best. Gray triangle indicates difference in WAIC scores.

C) Posterior distribution for the weight indicating the influence of RPE on drift rate, shaded red to indicate a meaningful effect (posterior mean=0.046, 95% HDI = [0.025, 0.067]).

I think modelling should provide clarity on relationships which cannot be revealed by simple statistical analyses.

We agree with the Reviewer and apologize for any confusion in the previous resubmission. Our DDM analysis indeed reveals a relationship which would be impossible to identify without modeling since both variables involved (RPE and drift rate) are latent (unobserved). We have provided additional interpretation of these relationships in response to the query above (now in Lines 40-42, 190-196). In addition, we now also test several non-linear alternative DDMs in response to another query above (Lines 124-126). However, if the Reviewer had alternative models in mind, it would be helpful if they can specify these alternative approaches. We would be more than happy to try them (within project's scope).

Also, rpe and PM can easily influence other parameters of the drift model, easiest prediction would be the non-decision time which might truncate for higher RPE or PM. I really don't want to create additional obstacles/burden to the authors, but I hope they can also see that uncorrected relationship testing here remains bit problematic.

We specifically focused on drift rate for theory-driven reasons: it is thought to capture the trial-varying quality of information associated with a cue (e.g. RPE) that enables a quick match to stimuli in memory. We now make this clearer on Lines 190-196 as below:

While the strength of visual information is implicated in evidence accumulation in perceptual decision-making (Palmer et al., 2005) our results suggest that the reward computations associated with surprising rewards provide more important evidence per unit of time for matching the recognition cue stimulus to the image stored in memory. This finding provides support for a functional dissociation between RPE- and perceptually-mediated memory enhancement.

Additionally, we appreciate the Reviewer's suggestions for additional DDM models which may provide more explanation of how RPE and PM modulate recognition memory choices, and tested the prediction that RPE and PM might be inversely correlated with non-decision time (ndt). Instead, we found that our original model examining drift rate significantly outperformed this alternative model (Review Figure 2, below). This comparison shows that the model utilizing non-decision time instead of drift rate appeared to be poorly specified, and not able to converge on stable posterior parameter distributions. As such, we chose not to include this alternative model in the manuscript.

Review Figure 1: Model comparison of two DDM models showing that the model assessing the linear influence of RPE and PM fits the behavioral data better than the models assessing the logarithmic or polynomial influence of these features. Gray triangle indicates difference in elpd WAIC scores.

I really want to leave it to the authors here, as it is a late stage in the revision process. They might want to construct/test few non-additive hypotheses to unpack how rpe and pm (and perhaps their

interaction) influence recognition decisions, or they can also write a detailed limitations sections overlapping with some of the issues that I highlighted above. I hope the authors find my comments useful.

We thank the Reviewer for their constructive commentary, which led us to add a computational modeling approach (DDM) to the memory portion of the analysis that we believe strengthens the study's findings and novelty. As described in our responses above, we now test several alternative DDMs to explore non-additive relationships (e.g. polynomial and logarithmic) between drift rate and RPE and PM, as well as an additional DDM which unpacks how RPE and PM modulate another DDM parameter, non-decision time.

3rd May 24

Dear Dr Qasim,

Your manuscript titled "Positive affect modulates memory by regulating the influence of reward prediction errors" has now been seen by our reviewers, whose comments appear below. In light of their advice I am delighted to say that we are happy, in principle, to publish a suitably revised version in Communications Psychology under the open access CC BY license (Creative Commons Attribution v4.0 International License).

We therefore invite you to revise your paper one last time to address the remaining editorial concerns requests. At the same time we ask that you edit your manuscript to comply with our format requirements and to maximise the accessibility and therefore the impact of your work.

EDITORIAL REQUESTS:

As a high-level point, I'll draw your attention to the fact that there is presently insufficient positive evidence for the absence of an effect of affect on the relationship between perceptual memorability and memory. This will need to be addressed. More information is enclosed in the attached "Editorial Requests Table". The Table also contains a comprehensive list of other requests that need to be resolved.

The table is similar but not identical to the one you kindly used to prepare your manuscript for re-review, so please pay close attention to each item, as these are now all critical to move to acceptance of the work. Please outline your response to each request in the right hand column. Please upload the completed table with your manuscript files as a Related Manuscript file.

SUBMISSION INFORMATION:

OPEN ACCESS:

Communications Psychology is a fully open access journal. Articles are made freely accessible on publication under a CC BY license (Creative Commons Attribution 4.0 International License). This license allows maximum dissemination and re-use of open access materials and is preferred by many research funding bodies.

For further information about article processing charges, open access funding, and advice and support from Nature Research, please visit <https://www.nature.com/commspsychol/article-processing-charges>

At acceptance, you will be provided with instructions for completing this CC BY license on behalf of all authors. This grants us the necessary permissions to publish your paper. Additionally, you will be asked to declare that all required third party permissions have been obtained, and to provide billing information in order to pay the article-processing charge (APC).

* **DATA AVAILABILITY:**

[link redacted]

Best regards,

Marike

on behalf of Patricia Lockwood

Patricia Lockwood, PhD

Editorial Board Member

Communications Psychology

Marike Schiffer, PhD

Chief Editor

Communications Psychology

REVIEWERS' COMMENTS:

Reviewer #1 (Remarks to the Author):

No further comments.

Reviewer #2 (Remarks to the Author):

The authors have now addressed all my remaining comments and the manuscript has greatly improved in clarity and quality as a result.

Reviewer #3 (Remarks to the Author):

I thank the authors for making further revisions based on my comments. I do not have any further recommendations